# The effect of transcutaneous auricular vagus nerve stimulation on cardiovascular function in subarachnoid hemorrhage patients: A randomized trial

Gansheng Tan[1,2†], Anna L Huguenard[1†], Kara M Donovan[1,2], Phillip Demarest[1,2], Xiaoxuan Liu[1,2], Ziwei Li[1,2], Markus Adamek[3], Kory Lavine[1], Ananth K Vellimana[1,4], Terrance T Kummer[4], Joshua W Osbun[1,4], Gregory J Zipfel[1], Peter Brunner[1,2], Eric C Leuthardt[1,2]*

[1]Department of Neurosurgery, Washington University School of Medicine, Springfield, United States; [2]Department of Biomedical Engineering, Washington University in St. Louis, St Louis, United States; [3]Department of Neuroscience, Washington University in St. Louis, St Louis, United States; [4]Department of Neurology, Washington University in St. Louis, St Louis, United States

*For correspondence: leuthardte@wustl.edu

†These authors contributed equally to this work

## eLife Assessment

The authors provide a **solid** set of data supporting the safety of transcutaneous auricular vagal nerve stimulation on cardiovascular parameters in the acute setting of critically ill patients presenting with subarachnoid hemorrhage. This **important** study also suggests a promising effect on autonomic balance.

## Abstract

**Background:** Subarachnoid hemorrhage (SAH) is characterized by intense central inflammation, leading to substantial post-hemorrhagic complications such as vasospasm and delayed cerebral ischemia. Given the anti-inflammatory effect of transcutaneous auricular vagus nerve stimulation (taVNS) and its ability to promote brain plasticity, taVNS has emerged as a promising therapeutic option for SAH patients. However, the effects of taVNS on cardiovascular dynamics in critically ill patients, like those with SAH, have not yet been investigated. Given the association between cardiac complications and elevated risk of poor clinical outcomes after SAH, it is essential to characterize the cardiovascular effects of taVNS to ensure this approach is safe in this fragile population. Therefore, this study assessed the impact of both acute and repetitive taVNS on cardiovascular function.

**Methods:** In this randomized clinical trial, 24 SAH patients were assigned to either a taVNS treatment or a sham treatment group. During their stay in the intensive care unit, we monitored patient electrocardiogram readings and vital signs. We compared long-term changes in heart rate, heart rate variability (HRV), QT interval, and blood pressure between the two groups. Additionally, we assessed the effects of acute taVNS by comparing cardiovascular metrics before, during, and after the intervention. We also explored acute cardiovascular biomarkers in patients exhibiting clinical improvement.

**Results:** We found that repetitive taVNS did not significantly alter heart rate, QT interval, blood pressure, or intracranial pressure (ICP). However, repetitive taVNS increased overall HRV and parasympathetic activity compared to the sham treatment. The increase in parasympathetic activity was most pronounced from 2 to 4 days after initial treatment (Cohen's $d$ = 0.50). Acutely, taVNS

increased heart rate, blood pressure, and peripheral perfusion index without affecting the corrected QT interval, ICP, or HRV. The acute post-treatment elevation in heart rate was more pronounced in patients who experienced a decrease of more than one point in their modified Rankin Score at the time of discharge.

**Conclusions:** Our study found that taVNS treatment did not induce adverse cardiovascular effects, such as bradycardia or QT prolongation, supporting its development as a safe immunomodulatory treatment approach for SAH patients. The observed acute increase in heart rate after taVNS treatment may serve as a biomarker for SAH patients who could derive greater benefit from this treatment.

**Funding:** The American Association of Neurological Surgeons (ALH), The Aneurysm and AVM Foundation (ALH), The National Institutes of Health R01-EB026439, P41-EB018783, U24-NS109103, R21-NS128307 (ECL, PB), McDonnell Center for Systems Neuroscience (ECL, PB), and Fondazione Neurone (PB).

**Clinical trial number:** NCT04557618.

## Introduction

Subarachnoid hemorrhage (SAH) is a devastating subtype of stroke that represents a significant global health burden and causes permanent disability in approximately 30% of survivors (**D'Souza, 2015**; **Weir, 2002**; **Zhang et al., 2023**). Early brain injury can occur within the first 24–48 hr after ictus, which involves a cascade of elevated intracranial pressure (ICP) and a subsequent drop of cerebral perfusion (**Schneider et al., 2018**). Systemic and local inflammation, cerebral edema, blood–brain barrier disruption, sympathetic nervous system activation, autoregulatory failure, microthrombosis, spreading depolarizations, and inflammation have all been observed during this period (**Macdonald et al., 2012**; **Budohoski et al., 2013**). These biological processes result in the inability of cerebral perfusion to match metabolic demands, leading to secondary brain injury and delayed cerebral ischemia that typically occurs between 5 and 14 days after the SAH (**Provencio, 2013**; **van Gijn et al., 2007**; **Tracey, 2002**; **Lv et al., 2018**). Delayed cerebral ischemia and deleterious inflammation are major predictors of poor outcomes and morbidity. The autonomic nervous system (ANS), comprising the sympathetic and the parasympathetic nervous system, plays a critical role in maintaining physiological homeostasis. SAH is believed to cause sympathetic predominance, which plays a key role in the development of cerebral vasospasm, renders patients more susceptible to non-neurological complications, and exacerbates deleterious inflammatory processes (**Naredi et al., 2000**).

Numerous interventions have been explored to address the complex pathologies of SAH that contribute to secondary brain injury, aiming to improve patient outcomes (**Provencio and Vora, 2005**). Transcutaneous auricular vagus nerve stimulation (taVNS) is one of the most promising therapeutic options, as recent studies have demonstrated its efficacy in reducing inflammation, improving autonomic balance, and enhancing brain plasticity (**Wu et al., 2023**; **Bonaz et al., 2016**; **Meyers et al., 2018**; **Provencio and Vora, 2005**; **Huguenard et al., 2023**). The auricular branch of the vagus nerve is a sensory nerve that innervates the external ear, including the cymba concha. Stimulating the auricular branch of the vagus nerve has been shown to activate the same brain regions as cervical vagus nerve stimulation (**Frangos et al., 2015**). Specifically, taVNS mediates cholinergic signaling and regulates pro-inflammatory responses via the inflammatory reflex (**Figure 1A**; **Sahn et al., 2023**; **Pavlov and Tracey, 2012**) In this reflex, inflammatory mediators such as cytokines trigger afferent vagus nerve signaling. This afferent signal then prompts an efferent response from the vagus nerve that acts to reduce the production of pro-inflammatory cytokines (**Tynan et al., 2022**).

The vagus nerve also mediates cardiovascular function by regulating the autonomic system and metabolic homeostasis (**Figure 1A**; **Keute et al., 2021**). Theoretically, taVNS increases parasympathetic activity, which can be measured as increased heart rate variability (HRV). While some animal studies have reported a potential risk of bradycardia and decreased blood pressure associated with vagus nerve stimulation, two reviews of human studies have considered the cardiovascular effects of taVNS generally safe, with adverse effects reported only in patients with pre-existing heart diseases (**Naggar et al., 2014**; **Hua et al., 2023**; **Kim et al., 2022**). However, its cardiovascular effect in SAH patients is largely unknown. Given that critically ill patients, such as those with SAH, are extremely vulnerable to cardiovascular complications, it is essential to thoroughly examine the cardiovascular

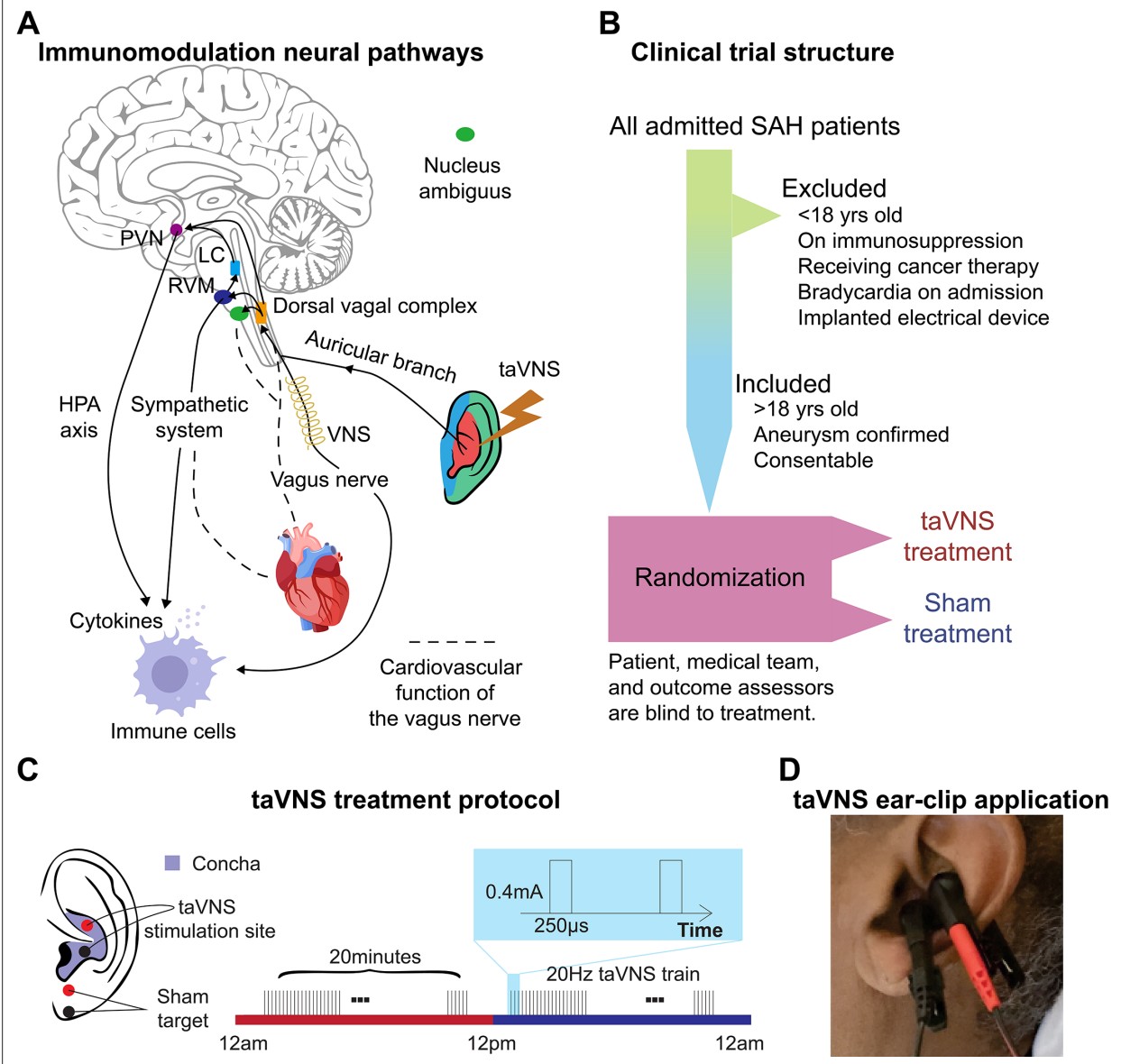

**Figure 1.** Study rationale and clinical trial design. (**A**) Immunomodulation neural pathways associated with vagus nerve stimulation include cholinergic anti-inflammatory pathway, sympathetic nervous system, and hypothalamic–pituitary–adrenal (HPA) axis. Immunogenic stimuli activate vagal afferents terminating primarily in the dorsal vagal complex. Ascending projections from the dorsal vagal complex reach the paraventricular nucleus (PVN) and rostral ventromedial medulla (RVM), activating the HPA axis and sympathetic system, respectively, to regulate the immune response. Transcutaneous auricular vagus nerve stimulation (taVNS) can affect cardiovascular function through the sympathetic system or efferent vagus nerve. (**B, C**) Clinical trial structure and treatment protocol. Patients in the taVNS group received electrical stimulation (0.4 mA, 250 μs pulse width, 20 Hz) for 20 min twice daily. Sham group patients wore the ear clip on the earlobe for the same duration. (**D**) Ear clip application for taVNS treatment.

implications of taVNS to ensure its safety in this fragile population. This is particularly notable as cardiovascular abnormalities following SAH, such as prolonged elevated heart rate and QT prolongation, are associated with an increased risk of poor outcomes (*Norberg et al., 2018*; *Schmidt et al., 2014*; *Zhang and Qi, 2016*; *van der Bilt et al., 2009*). However, our limited understanding of these effects constitutes a significant barrier, preventing the advancement of taVNS from a promising therapeutic approach to an established clinical treatment for SAH. To address this gap, we assessed the effects of acute and repetitive taVNS on cardiovascular function based on electrocardiogram (ECG) and other monitored vital signs from SAH patients in the intensive care unit (ICU). The current study is part of the NAVSaH trial (NCT04557618) and focuses on the trial's secondary outcomes, including heart rate, QT interval, HRV, and blood pressure (*Huguenard et al., 2024a*). This interim analysis aims

to evaluate the cardiovascular safety of the taVNS protocol and to provide insights that will inform the application of taVNS in SAH patients. The primary outcomes of this trial, including change in the inflammatory cytokine tumor necrosis factor-alpha (TNF-α) and rate of radiographic vasospasm, are available as a pre-print and currently under review (*Huguenard et al., 2024b*). Based on a meta-analysis, most aversive effects were seen in repeated sessions lasting 60 min or more; therefore, we hypothesized that repetitive taVNS increased HRV but did not cause bradycardia and QT prolongation (*Kim et al., 2022*). To test this hypothesis, we compared changes in cardiovascular metrics at the phase of early brain injury (within 72 hr) and at the phase when delayed cerebral ischemia develops (after Day 4) between patients receiving taVNS and sham treatments. Root mean square of successive differences (RMSSD) and the standard deviation of normal RR intervals (SDNN) are two commonly used HRV metrics. RMSSD indicates parasympathetic activity, while lower SDNN is associated with increased cardiac risk (*Kleiger et al., 2005*; *Shaffer and Ginsberg, 2017*). Providing effective taVNS treatment modulates the autonomic system, we propose that heart rate or HRV following acute taVNS could inform which SAH patients may experience the most clinical benefit from taVNS treatment. To explore this possibility, we correlated the changes in heart rate and HRV following acute taVNS

**Table 1.** Patient demography.
HH: Hunt & Hess classification. mRS: modified Rankin Scale. Y: yes. N: no.

| Patient | Decade of life | Gender | Race | HH | mRS (admission) | mRS (discharge) | Vasospasm treated by blood pressure goal augmentation | Indwelling arterial lines | ICP monitoring |
|---|---|---|---|---|---|---|---|---|---|
| 1 | 70s | M | White | 3 | 5 | 5 | Y | Y | Y |
| 2 | 50s | M | Black/African American | 3 | 4 | 3 | N | Y | Y |
| 3 | 50s | F | White | 2 | 2 | 3 | N | N | N |
| 4 | 60s | M | White | 2 | 3 | 4 | Y | N | N |
| 5 | 60s | F | White | 2 | 2 | 3 | Y | N | N |
| 6 | 70s | F | White | 2 | 3 | 2 | N | Y | N |
| 7 | 60s | M | White | 4 | 5 | 4 | N | Y | Y |
| 8 | 60s | F | White | 4 | 5 | 2 | N | Y | Y |
| 9 | 70s | F | White | 2 | 3 | 0 | Y | N | N |
| 10 | 70s | F | White | 2 | 3 | 3 | Y | Y | N |
| 11 | 40s | M | Black/African American | 4 | 5 | 4 | Y | Y | Y |
| 12 | 40s | F | White | 2 | 4 | 3 | N | N | N |
| 13 | 50s | F | Black/African American | 2 | 2 | 1 | Y | N | N |
| 14 | 80s | F | White | 1 | 3 | 3 | Y | N | N |
| 15 | 70s | F | Black/African American | 4 | 5 | 3 | N | N | Y |
| 16 | 40s | F | Black/African American | 2 | 2 | 3 | N | N | N |
| 17 | 60s | F | White | 2 | 1 | 0 | N | N | N |
| 18 | 30s | F | White | 3 | 4 | 2 | N | N | Y |
| 19 | 60s | F | White | 3 | 4 | 2 | N | N | Y |
| 20 | 80s | F | White | 2 | 2 | 3 | N | Y | Y |
| 21 | 50s | F | White | 3 | 4 | 4 | N | Y | Y |
| 22 | 70s | F | White | 1 | 2 | 4 | N | N | N |
| 23 | 40s | F | Black/African American | 2 | 2 | 4 | N | Y | N |
| 24 | 60s | M | White | 4 | 5 | 5 | Y | Y | Y |

treatment and changes in the modified Rankin Score (mRS), which measures the degree of disability or dependence in the daily activities of people suffering from neurological disability.

## Results

Twenty-four participants were randomized to receive the taVNS ($N$ = 11) or Sham ($N$ = 13) treatment (*Table 1*, *Figure 2—figure supplement 2*). *Appendix 3—table 1* shows the clinical characteristics of the two treatment groups. The participants, the medical team who dictated all management decisions for the patient's SAH, and the outcomes assessors who assigned mRS at admission and discharge were blinded to the treatment. The structure of this study is shown in *Figure 1B*. Following randomization, enrolled patients underwent 20 min of either taVNS or sham stimulation twice daily during their stay in the ICU. This treatment schedule was informed by findings from Addorisio et al., where a 5-min taVNS protocol was administered twice daily to patients with rheumatoid arthritis for 2 days (*Addorisio et al., 2019*). Their study found that circulating C-reactive protein levels significantly reduced after 2 days of treatment but returned to baseline at the second clinical assessment by Day 7. Given the high inflammatory state associated with SAH and our intention to maintain a steady reduction in inflammation, we decided to extend the treatment duration to 20 min per session. During treatment periods, a portable transcutaneous electrical nerve stimulation (TENS) device (TENS 7000 Digital TENS Unit, Compass Health Brands, OH, USA) was connected to the patient's left ear using two ear clips (*Figure 1C, D*). For taVNS treatments, these ear clips were placed along the concha of the ear, while for sham treatments, the clips were placed along the earlobe to avoid stimulation of the auricular vagus nerve from tactile pressure (*Figure 1C*). For the taVNS group, stimulation parameters were selected based on values reported in prior studies that sought to maximize vagus somatosensory evoked potentials while avoiding the perception of pain: 20 Hz frequency, 250 µs pulse width, and 0.4 mA intensity (*de Gurtubay et al., 2021*). The stimulation parameters were designed to be imperceptible to the patients, and there were no reports of detection of taVNS, suggesting the success of the blinding. No electrical current was delivered during sham treatments. Please see *Huguenard et al., 2024a* for a detailed protocol of this study.

### Effects of repetitive taVNS on cardiac function

A study has shown that 15 min of taVNS reduced sympathetic activity in healthy individuals, with effects that persist during the recovery period (*Clancy et al., 2014*). This finding suggests that taVNS may exert long-term effects on cardiovascular function. Therefore, we investigated whether repetitive taVNS treatment affects heart rate and QT interval, key indicators of bradycardia or QT prolongation, using 24-hr ECG recording. We found no significant differences in heart rate between groups (*Figure 2D*, Mann–Whitney $U$ test, $N$(taVNS) = 94, $N$(Sham) = 95, p-value = 0.69, Cohen's $d$ = −0.01, $W$-statistics = 4317, power = 0.93). Changes in heart rate from Day 1 were equivalent between groups (two-tailed equivalence tests, p[lower threshold] = 0.006, test statistics[lower threshold] = 2.53; p[lower threshold] = 0.004, test statistics[lower threshold] = −2.72, $N$(VNS) = 94, $N$(VNS) = 95). We further confirmed that changes in heart rate were similar between treatment groups following SAH (*Figure 2D*, |Cohen's $d$|<0.2 for Days 2–4, 5–8, 8–10, and 11–13). Moreover, changes in corrected QT interval from Day 1 were significantly higher in the Sham group compared to the taVNS group (*Figure 2E*, Mann–Whitney $U$ test, $N$(taVNS) = 94, $N$(Sham) = 95, p-value <0.001, Cohen's $d$ = −0.57). Similarly, uncorrected QT intervals from Day 1 were higher in the Sham group (*Figure 2—figure supplement 1*, Cohen's $d$ = −0.42). After the phase of early brain injury, the mean and median corrected QT interval were lower in the taVNS group with large effect sizes (*Figure 2E*, |Cohen's $d$| >0.5). To ensure that repetitive taVNS did not lead to QT prolongation outside the stimulation period, we calculated the percentage of prolonged QT intervals. Prolonged QT intervals were defined as corrected QT interval ≥500 ms. We found that changes in prolonged QT intervals percentage from Day 1 were higher in the Sham group (*Figure 2F*, Mann–Whitney $U$ test, $N$(taVNS) = 94, $N$(Sham) = 95, p-value <0.001, Cohen's $d$ = −0.72).

Subsequently, we investigated the effect of taVNS treatment on RMSSD and SDNN. We found that changes in SDNN using Day 1 as baseline were not significantly different between the treatment groups (*T*-test, $N$(taVNS) = 94, $N$(Sham) = 95, p = 0.479, Cohen's $d$ = 0.10, $t$ statistics = 0.71, *Figure 3A*). Changes in RMSSD using Day 1 as baseline were significantly higher in the taVNS treatment group (*T*-test, $N$(taVNS) = 94, $N$(Sham) = 95, Bonferroni-corrected p = 0.025, Cohen's $d$ = 0.42,

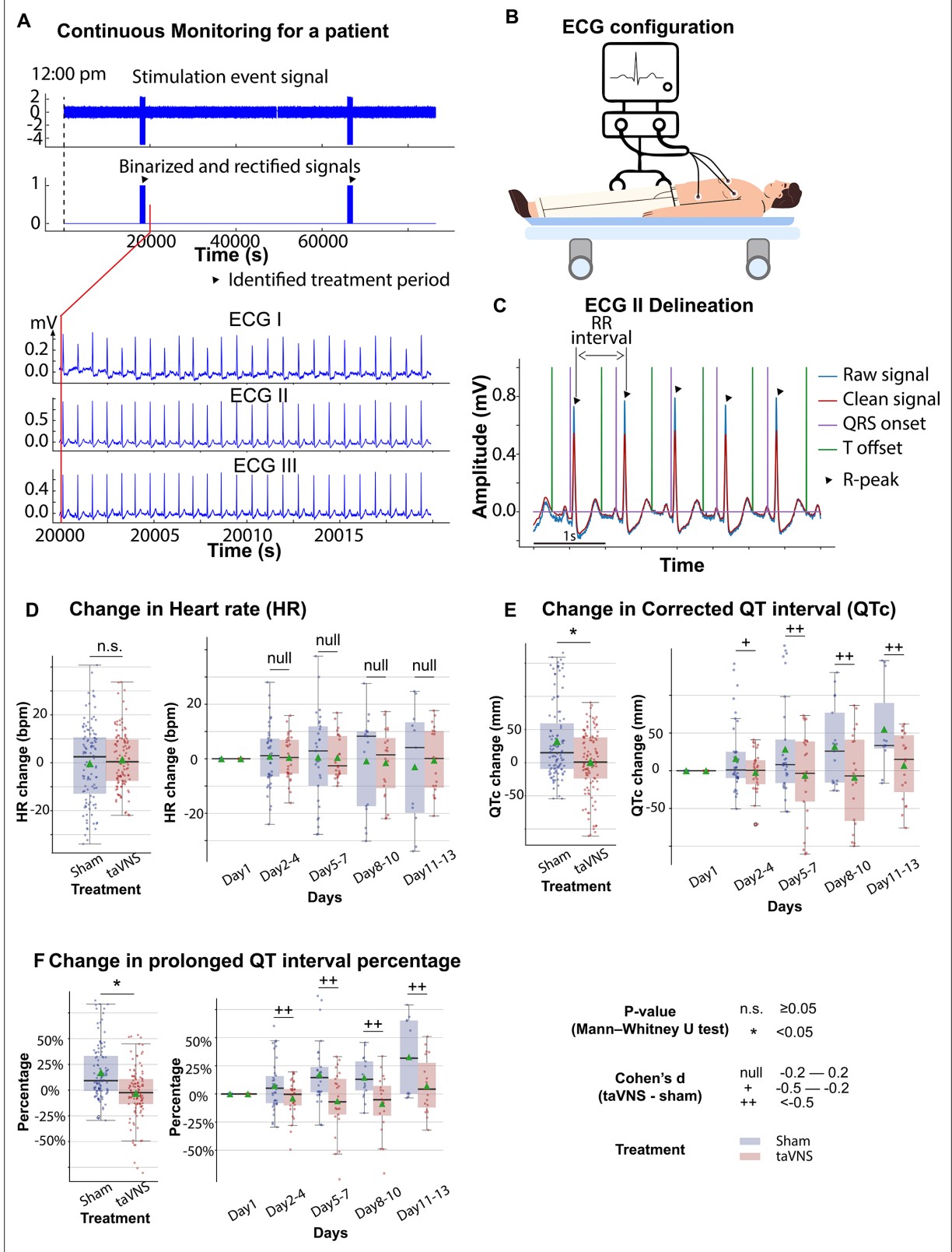

**Figure 2.** The effects of transcutaneous auricular vagus nerve stimulation (taVNS) on cardiac function. (**A**) Signals encoding treatment period and electrocardiogram (ECG) signals in a representative patient. (**B**) 3-lead ECG configuration in the intensive care unit. (**C**) P wave, T wave, and QRS complex are delineated from clean ECG II signals. (**D, E**) Heart rate and corrected QT interval changes from the first hospitalized day in the two treatment groups. (**F**) Changes in the percentage of prolonged QT from the first hospitalized day in the two treatment groups.

*Figure 2 continued on next page*

*Figure 2 continued*

The online version of this article includes the following figure supplement(s) for figure 2:

**Figure supplement 1.** The effect of repetitive and acute transcutaneous auricular vagus nerve stimulation (taVNS) on uncorrected QT interval.

**Figure supplement 2.** The distribution of Hunt & Hess classification and modified Rankin Scale for transcutaneous auricular vagus nerve stimulation (taVNS) and Sham groups.

*t* statistics = 2.91, *Figure 3B*). We further studied the effects of taVNS in different phases following SAH. *Figure 3A-B* show the changes in SDNN and RMSSD in bins of 3 days for the two treatment groups. The taVNS treatment increased RMSSD over the course of the treatment, with a smaller effect size (Cohen's *d* = 0.29) observed between Days 2 and 4, corresponding to the early brain injury phase, and large effect sizes at the later phases (Cohen's *d* = 0.41 for Days 5–7, Cohen's *d* = 0.54 for Days 8–10, Cohen's *d* = 0.66 for Days 11–13). We further tested if the RMSSD reduction rate was greater in the Sham treatment group with a linear regression model: RMSSD change ~Day * Treatment. The results show that the RMSSD reduced slower in the taVNS treatment group when compared to the sham treatment, but this trend did not reach significance (coefficient of taVNS * Day interaction effect = 2.00, p = 0.21, *Figure 3—figure supplement 1*).

RMSSD and SDNN are two of the most commonly used methods for quantifying HRV. Bartlett's test indicated that there are significant correlations among these measures (p < 0.01, *Figure 3C*). To analyze the effect of taVNS treatment on the autonomic system, we used factor analysis to identify the underlying factors. We focused on the two factors with the greatest eigenvalue. *Figure 3D* shows the factor loading, that is, the variance explained by HRV metrics on the two factors. The first factor correlates positively with metrics representing variability, including RMSSD, SDNN, pNNI_50, and total power, and therefore has been termed overall HRV. The second factor correlated positively with RMSSD and normalized high-frequency power, representing parasympathetic activity, and negatively correlated with the cardiac sympathetic index (CSI). Hence, it is termed parasympathetic activity. Overall HRV change from Day 1 was significantly higher in the VNS group (*Figure 3E*, Mann–Whitney *U* test, *N*(taVNS) = 94, *N*(Sham) = 95, p-value = 0.04, Cohen's *d* = 0.37). The effect size was trivial between Days 2 and 4 and increased over the course of treatment. The parasympathetic activity was also significantly higher in the VNS treatment group, and we observed the largest effect size between Days 2 and 4 (Cohen's *d* = 0.50, *Figure 3F*).

We also investigated the potential association between clinical outcomes, as measured by changes in the mRS from admission to discharge, and HRV metrics. We found that heart rate was lower in patients with improved mRS (i.e., <0) (Mann–Whitney *U* test, *N*(mRS < 0) = 122, *N*(mRS > 0) = 98, p-value <0.01, Cohen's *d* = −0.54). Parasympathetic activity, overall HRV, and corrected QT interval did not differ significantly between patients with improved mRS and patients with worsened mRS (*Figure 3—figure supplement 4*).

## Effects of repetitive taVNS on vascular function

Elevated blood pressure is a common occurrence in SAH and is linked with a higher risk of re-rupture of cerebral aneurysms and vasospasm (*Minhas et al., 2022*; *Hosmann et al., 2020*). In this study, patients in both treatment groups received medical treatment determined by the medical team, including vasopressors and medication for blood pressure management. We investigated whether taVNS induced any additional blood pressure changes beyond those managed by the medical team. We found that the median and mean blood pressure change from the first hospitalized day were greater than 0 for both treatment groups (*Figure 4B*). No significant differences were detected in changes in blood pressure and ICP between the treatment groups (*Figure 4B, C*). Equivalence testing confirmed that the ICP changes from the first hospitalization day were not significantly different between treatment groups, with a 2-mmHg equivalence margin (two one-sided *t*-tests, p[lower threshold] = 3.66 × 10$^{-13}$, *t*[lower threshold] = 8.07; p[upper threshold] = 3.33 × 10$^{-10}$, *t*[upper threshold] = −6.73). Equivalence testing also indicated that there were no significant different changes in blood pressure between treatment groups (two one-sided *t*-tests, p[lower threshold] = 0.07, *t*[lower threshold] = 1.51; p[upper threshold] = 0.002, *t*[upper threshold] = −3.00). We further verified that there were no significant changes in arterial line blood pressure obtained via continuous invasive monitoring between treatment groups (*Figure 4—figure supplement 1*). We subsequently compared the plethysmography

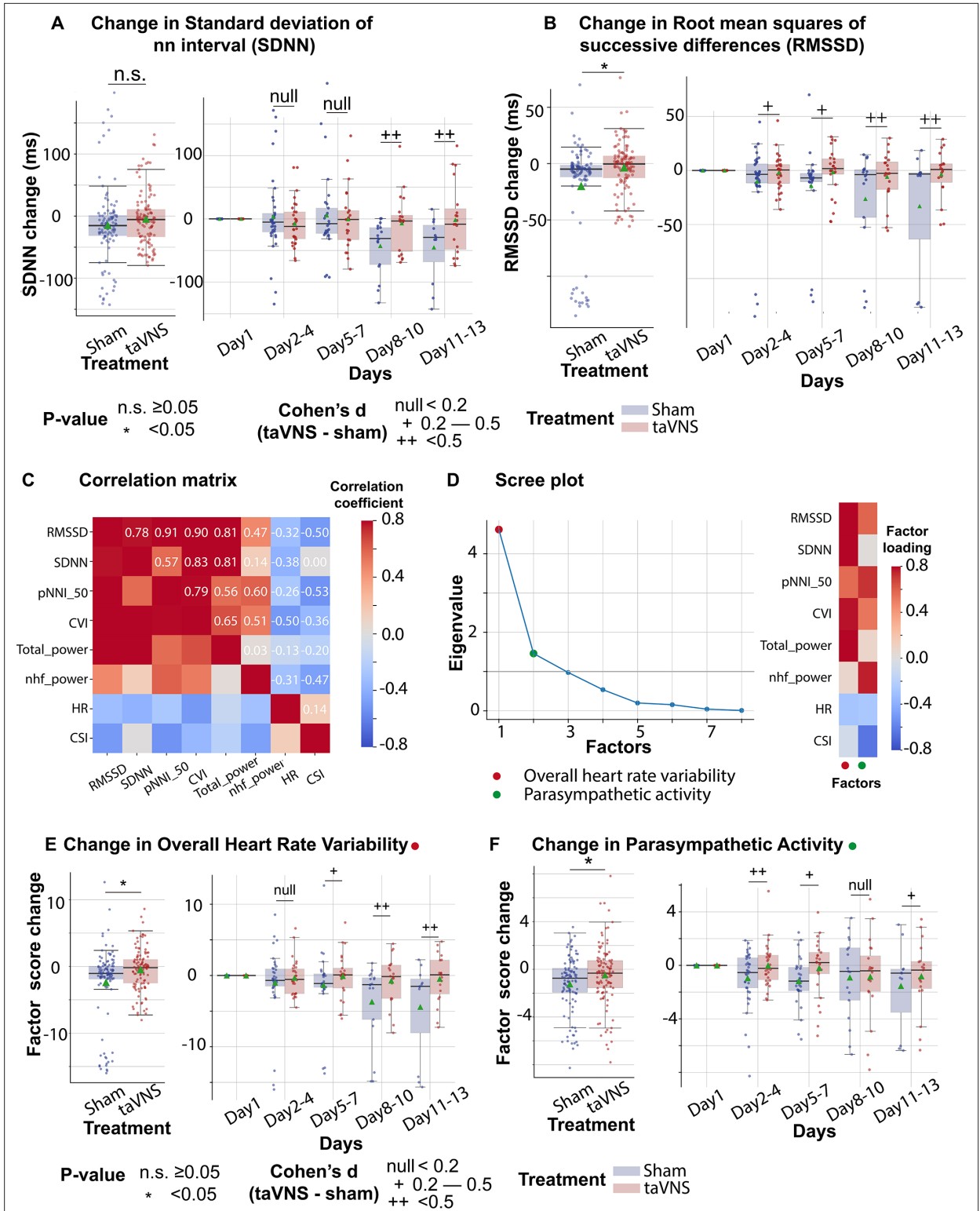

**Figure 3.** The effects of transcutaneous auricular vagus nerve stimulation (taVNS) on overall heart rate variability and parasympathetic activity. (**A, B**) Changes in standard deviation of NN interval (SDNN) changes and root mean squares of successive differences over time for the two treatment groups. The color represents the treatment group. Green triangles represent the mean. (**C**) Correlation between standard electrocardiogram (ECG) features underlying autonomic activities. (**D**) Factor analysis showed that there are two factors underlying the standard ECG features. The first factor is referred to as overall heart rate variability. The second factor is referred to as parasympathetic activity. (**E, F**) The effect of taVNS on the two factors. pNNI_50:

*Figure 3 continued on next page*

*Figure 3 continued*

percentage of number of successive NN intervals that differ by more than 50 ms. CVI: cardiac vagal index. Total power: total power below 0.4 Hz of normal RR interval. nhf_power: relative power of the high-frequency band (0.15–0.4 Hz). CSI: cardiac sympathetic index.

The online version of this article includes the following figure supplement(s) for figure 3:

**Figure supplement 1.** Heart rate variability in transcutaneous auricular vagus nerve stimulation (taVNS) and Sham treatment groups.

**Figure supplement 2.** Time- and frequency-domain cardiac measures in transcutaneous auricular vagus nerve stimulation (taVNS) and Sham treatment groups.

**Figure supplement 3.** The normalized high-frequency power may not fully represent parasympathetic activity when the respiration rate exceeds 25 bpm.

**Figure supplement 4.** The impact of clinical outcome on heart rate variability.

peripheral perfusion index (PPI) between the treatment groups as it is a proxy metric for cardiac stroke volume and vascular tone (*Elshal et al., 2021*; *Coutrot et al., 2021*). We found that PPI change was significantly lower (Mann–Whitney $U$ test, $N$(taVNS) = 83, $N$(Sham) = 95, Bonferroni corrected p < 0.01, Cohen's $d$ = −0.49). In addition, respiration rate change was significantly higher (Mann–Whitney $U$ test, $N$(taVNS) = 94, $N$(Sham) = 95, Bonferroni corrected p = 0.02, Cohen's $d$ = 0.37) in the taVNS group, as compared to the Sham group (*Figure 4D, E*). We hypothesized that the increase in respiratory rate was a compensatory mechanism to ensure similar oxygen delivery. We found a significant negative correlation between changes in PPI and changes in respiration rate only for the taVNS treatment group (Pearson correlation coefficient = −0.37, p < 0.001, $t$-test, *Figure 4—figure supplement 1D*). The Pearson correlation coefficient for the sham treatment group is −0.08 (p = 0.36).

## Acute effects of taVNS on cardiovascular function

Assessing the acute effect of taVNS on cardiovascular is crucial for its safe translation into clinical practice. We compared the acute change of heart rate, corrected QT interval, and HRV between treatment groups, as abrupt changes in the pacing cycle may increase the risk of arrhythmias (*Zaniboni, 2024*). The change in heart rate from treatment onset is shown in *Figure 5B*. We subsequently tested whether taVNS affects changes in heart rate between post- and pre-treatment. We found that the changes in heart rate were not significantly different between treatment groups although heart rate increased in the taVNS group (Wilcoxon rank-sum test, $N$ = 188, Bonferroni corrected p = 0.03, Cohen's $d$ = 0.11) but not in the Sham group (Wilcoxon signed-rank test, $N$ = 199, Bonferroni corrected p = 0.72, Cohen's $d$ = 0.00) (*Figure 5C*). However, the increase in heart rate after taVNS was within 0.5 standard deviations of daily heart rate. There were no significant differences in changes in corrected QT interval or HRV, as measured by RMSSD, SDNN, and relative power of high-frequency band between treatment groups (*Figure 5D, E*, *Figure 5—figure supplements 1 and 3*). *Figure 2—figure supplement 1C* shows the acute changes in uncorrected QT interval. *Appendix 3—table 1* summarizes the absolute changes in cardiovascular metrics for the treatment groups. We further asked whether heart rate can serve as a biomarker that indicates which SAH patients would receive the greatest benefit from continuing taVNS treatment. We investigated the relationship between changes in heart rate from pre- to post-taVNS treatment and changes in mRS between admission and discharge using a linear mixed-effects model. In this model, the treatment group, mRS change, and their interaction were included as fixed effects, while subject was included as a random effect. Our analysis revealed that the slope between changes in heart rate and changes in mRS is significantly more negative for the taVNS treatment group (*Table 2*). This finding suggests that an increase in heart rate following acute taVNS treatment is associated with improved clinical outcomes (*Figure 5F*). Post hoc analysis showed that patients in the taVNS treatment group who had an improvement in mRS of –2 or greater compared to other patients had significantly greater increases in heart rate (Mann–Whitney $U$ test, p = 0.02, $N$(mRS change < −1) = 53, $N$(mRS change ≥ −1) = 135, Cohen's $d$ = 0.34, *Figure 5—figure supplement 1C*). Conversely, HRV change, represented by RMSSD, was not significantly different based on mRS in SAH patients (*Figure 5—figure supplement 1D*).

Subsequently, we compared changes in blood pressure, PPI, ICP, and respiration rate from pre- to post-treatment periods between treatment groups. We found that changes in PPI and blood pressure were significantly higher in the taVNS group, as compared to the Sham group (Mann–Whitney $U$ test, blood pressure: p = 0.03, Cohen's $d$ = 0.22, $N$ = 180 for Sham and 159 for taVNS; PPI: p <

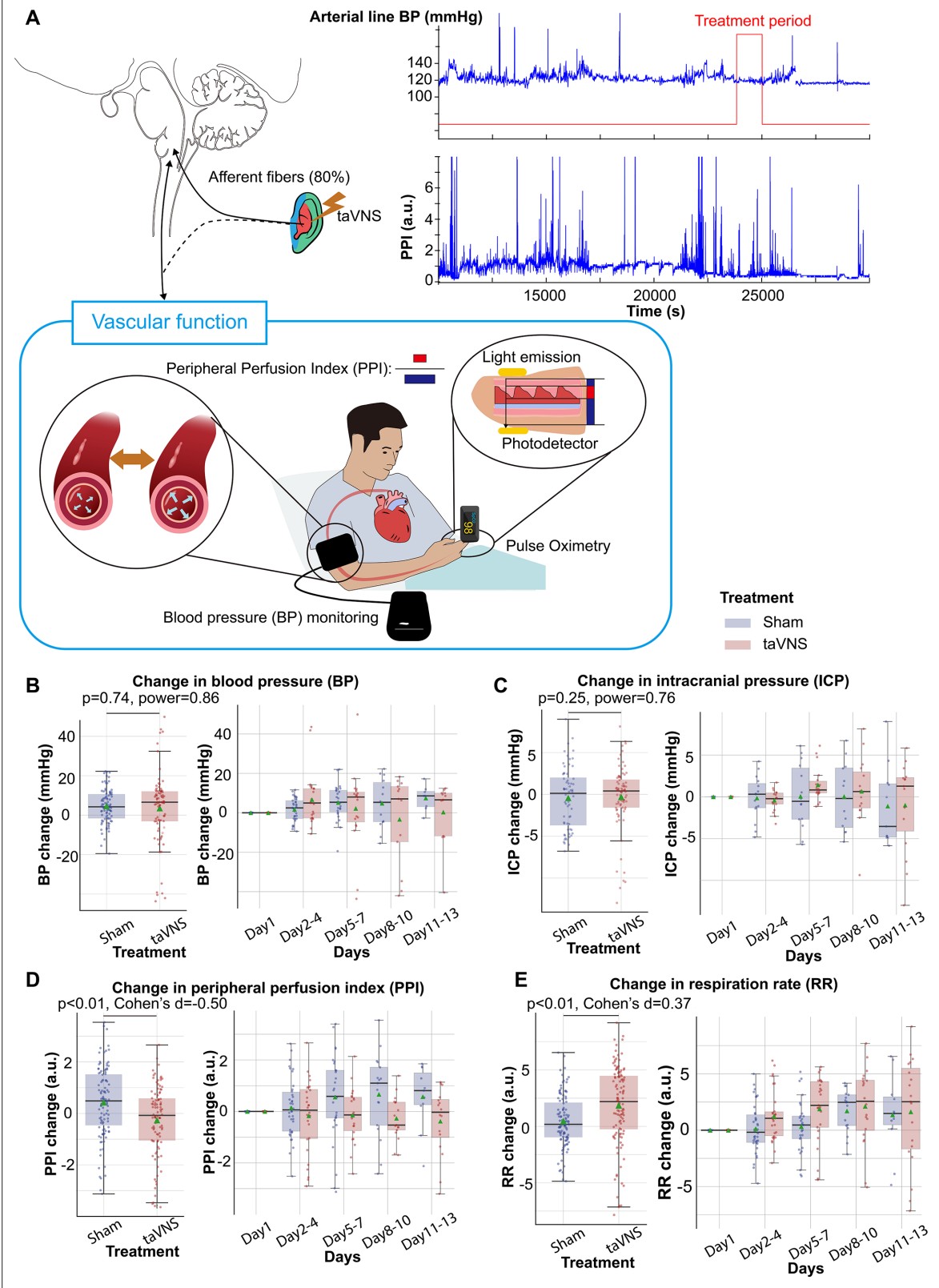

**Figure 4.** Effects of repetitive transcutaneous auricular vagus nerve stimulation (taVNS) on vascular function. (**A**) Representative vital signs and their physiology. Arterial line blood pressure (see ***Figure 4—figure supplement 1***), intracranial pressure (ICP), and mean blood pressure measured regularly by nurses (BP) were recorded. Blood pressure is an index of vasodilation. Peripheral perfusion index (PPI) is the ratio between the pulsatile and the non-

*Figure 4 continued on next page*

*Figure 4 continued*

pulsatile blood flow, reflecting the cardiac output. (**B, C**) Mean BP and ICP changes from the first hospitalization day did not differ significantly between the treatment groups. (**D, E**) PPI change from the first hospitalized day was lower in the taVNS treatment group, while RR change was higher.

The online version of this article includes the following figure supplement(s) for figure 4:

**Figure supplement 1.** The effect of transcutaneous auricular vagus nerve stimulation (taVNS) on arterial line blood pressure monitoring and noninvasive blood pressure monitoring.

0.01, Cohen's *d* = 0.19, *N* = 227 for Sham and 186 for taVNS, *Figure 5—figure supplement 2*). Only PPI remained significantly different between treatment groups after Bonferroni correction. The acute changes in PPI and blood pressure remained within the daily standard deviation. No significant differences in post-treatment changes in ICP or respiration rate were observed between treatment groups.

## Discussion

This study examined the effects of taVNS on cardiovascular function in patients with SAH. We investigated both the cumulative and acute impacts of taVNS. The findings in our study indicate that repetitive taVNS is not associated with previously suggested risks, such as bradycardia and QT prolongation. Furthermore, repetitive taVNS treatment increased overall HRV and parasympathetic activity, which are indicators of a healthy cardiovascular system. When looking at the acute effects, taVNS only significantly increased the PPI but not heart rate, HRV, corrected QT interval, blood pressure, or ICP. The findings are summarized in *Table 3*. Interestingly, we found that heart rate can serve as a biomarker for identifying SAH patients who are most likely to benefit from taVNS treatment. Collectively, this study substantiates the safety of treating SAH patients with taVNS and provides foundational data for future efforts to optimize and translate taVNS therapy toward clinical use.

### taVNS and autonomic system

The ANS, comprising the sympathetic and the parasympathetic nervous system, plays a critical role in maintaining physiological homeostasis. These two systems work synergistically to mediate interactions between the nervous and immune systems, which is thought to be the underlying mechanism for the immunomodulatory effect of taVNS. Our study is aligned with the finding that the autonomic balance is shifted toward sympathetic dominance following SAH (*Figure 3*, *Figure 3—figure supplement 2*; *Chiu et al., 2012*; *Bai et al., 2023*). In addition, we found that dysregulation of sympathovagal balance toward sympathetic dominance could be restored by taVNS treatment.

A key metric that reflects this restored sympathovagal balance is HRV (*Figure 3F*). Specifically, factor analysis based on HRV metrics showed that the parasympathetic activity was significantly higher in the taVNS treatment group. This difference was most pronounced during the early phase, between Days 2 and 4 following SAH. In addition to analyzing the correlation between the parasympathetic activity factor and established HRV measures that reflect parasympathetic activity, such as RMSSD and pNNI_50 (*Figure 3C*), we also examined changes in a frequency-domain HRV measure—the relative power of the high-frequency band (0.15–0.4 Hz)—to validate the accuracy of the factor analysis. The relative power of the high-frequency band is widely used to indicate respiratory sinus arrhythmia, a process primarily driven by the parasympathetic nervous system. We found that both the change in parasympathetic activity factor and relative high-frequency power were higher in the taVNS group at the early phase (Days 2–4, *Figure 3—figure supplement 2*). Conversely, we observed higher high-frequency power in the Sham group during the later phase. If the factor analysis successfully isolates the parasympathetic activity, there should be other factors than the parasympathetic activity affecting the relative power of the high-frequency band. One such factor is the respiration rate. The high-frequency range is between 0.15 and 0.4 Hz, corresponding to respiration's frequency range of approximately 9–24 breaths per minute. If the respiration rate increases and exceeds 24 breaths per minute, the respiratory-driven HRV might occur at a frequency higher than the typical high-frequency band. Given that the respiration rate was higher in the taVNS treatment group, a compensatory mechanism to ensure oxygen delivery (*Figure 4E*), we hypothesized that the observed lower high-frequency power in the taVNS treatment group compared to sham at later phases was a result of increased respiration rate. Indeed, we found the normalized high-frequency power was higher when RR was less than 25 bpm compared to when RR >25 bpm (Cohen's *d* = 0.85, *Figure 3—figure supplement*

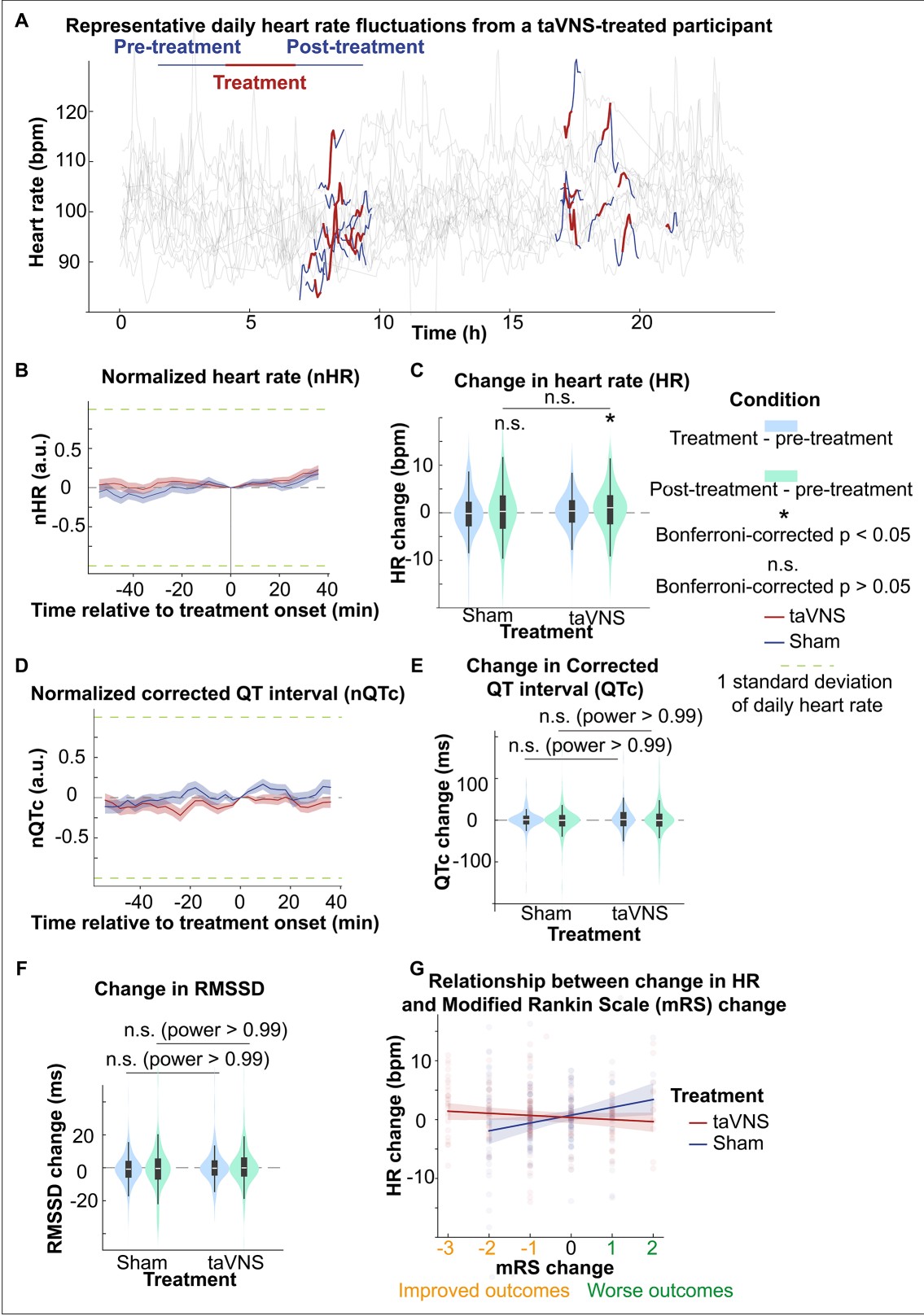

**Figure 5.** The acute effects of transcutaneous auricular vagus nerve stimulation (taVNS) on cardiac function. (**A**) Daily fluctuation of heart rate of a subject receiving VNS treatment. The treatment period, a 20-min period before and after treatment, is highlighted. Note that a small proportion of electrocardiogram (ECG) signals to derive heart rate was missing due to the expected cyclical restarting of the monitoring system. (**B, D**) Normalized heart rate (QTc) aligned at the treatment onset over time for the two treatment groups. The heart rate (QTc) is normalized based on the mean and

*Figure 5 continued on next page*

*Figure 5 continued*

standard error of heart rate for each day. (**C**) The difference in HR between the treatment period, post-treatment period, and pre-treatment period for the two groups. Wilcoxon signed-rank test was used to test if the HR difference is statistically different from 0 in the VNS treatment group. Bonferroni-corrected p-value for HR difference between post-treatment and treatment period is 0.03 ($N = 188$, Cohen's $d = 0.1$). Mann–Whitney $U$ tests were used to compare cardiac function metric differences between the two treatment groups. (**E, F**) The difference in QTc and RMSSD between the treatment period, post-treatment period, and pre-treatment period for the two groups (**G**) The relationship between heart rate changes following acute taVNS and functional outcome.

The online version of this article includes the following figure supplement(s) for figure 5:

**Figure supplement 1.** Cardiac effects of acute transcutaneous auricular vagus nerve stimulation (taVNS).

**Figure supplement 2.** Vascular effects of acute transcutaneous auricular vagus nerve stimulation (taVNS).

**Figure supplement 3.** The effect of acute transcutaneous auricular vagus nerve stimulation (taVNS) on peak frequencies within the high-frequency (HF) and the low-frequency (LF) bands.

*3A*). Moreover, an increase in RR in the taVNS treatment group was associated with a decrease in high-frequency power (*Figure 3—figure supplement 3B*). These control analyses underscored the necessity of performing factor analysis to robustly measure parasympathetic activities and confirm that taVNS treatment mitigated the sympathetic overactivation during the early phase.

Additionally, taVNS led to a decreased QTc without a significant change in heart rate, mimicking the effects observed with propranolol administration, a beta-blocker that reduces sympathetic activity (*Solti et al., 1989*). This finding suggests that repetitive taVNS reduces sympathetic overactivation and influences ventricular repolarization processes. Age affects sinus node function and is potentially associated with a higher risk of poor outcomes. To control for individual differences, including those due to age, our study compared the change in cardiovascular parameters from Day 1 within each subject across treatment groups. To further verify if age influences autonomic changes following SAH, we performed ANCOVA on autonomic function parameters with age included as a covariate. This analysis showed that age was negatively correlated with changes in heart rate, SDNN, and RMSSD from Day 1 but not with changes in QT intervals. After adjusting for age, we found that RMSSD and SDNN changes were significantly higher, while QTc changes were significantly lower in the taVNS treatment group (*Appendix 4—table 1*). These results align with the conclusion that repetitive taVNS treatment increased HRV and was unlikely to cause bradycardia or QT prolongation. In addition, autonomic changes following SAH may be influenced by age. Specifically, lower RMSSD and SDNN in older individuals suggest a greater shift toward sympathetic predominance following SAH (*Appendix 4—table 1*).

PPI is primarily influenced by cardiac output and vascular tone. Elevated PPI is associated with vasodilation and/or increased stroke volume. In the Sham group, increases in both PPI and blood pressure were observed when compared to Day 1 values (*Figure 3*). This effect may be due to higher stroke volume resulting from sympathetic activation following SAH. Alternatively, this could represent the heightened need for vasopressor interventions to improve cerebral perfusion due to more robust sympathetically driven cerebral vasospasm. The increase in PPI was less for the taVNS treatment group (*Figure 4*), suggesting a restored autonomic balance in the taVNS treatment group. However, the

**Table 2.** Relationship between HR changes following acute transcutaneous auricular vagus nerve stimulation (taVNS) and clinical outcomes.

| Model | $HR_{subject, treatment}(mRS) = \beta_0 + \beta_{taVNS} \times 1_{treatment=taVNS} + (\beta_{mRS} + \beta_{taVNS,slope} \times 1_{treatment=taVNS}) \times mRS + u_{subject}$ | | |
|---|---|---|---|
| | Coefficient | p-value | 95% confidence interval |
| $\beta_0$ | 0.73 | 0.211 | −0.41 to 1.87 |
| $\beta_{taVNS}$ | −0.29 | 0.737 | −1.95 to 1.38 |
| $\beta_{mRS}$ | 1.47 | **0.006** | 0.43 to 2.51 |
| $\beta_{taVNS,slope}$ | −1.85 | **0.005** | −3.12 to 0.57 |

Bold values denote statistical significance at the p < 0.05 level.

**Table 3.** Summary of effects of acute and repetitive transcutaneous auricular vagus nerve stimulation (taVNS) on cardiovascular function in subarachnoid hemorrhage (SAH) patients.
Metrics for cardiovascular function include heart rate variability, heart rate, QT interval, blood pressure, intracranial pressure, peripheral perfusion index, and respiration rate.

| | | Positive findings | Null findings | Implications |
|---|---|---|---|---|
| Repetitive | Cardiac function | Increased HRV | QT interval<br>Heart rate | Increased parasympathetic activity |
| | Vascular function | Reduced peripheral perfusion index<br>Increased respiration rate | Blood pressure<br>Intracranial pressure | Compensatory mechanisms to maintain autonomic balance |
| Acute | Cardiac function | | Heart rate<br>Heart rate variability<br>QT interval | No to small acute effect |
| | Vascular function | Increased peripheral perfusion index | Intracranial pressure<br>Respiration rate | |

effects of taVNS on blood pressure require further investigation as more than half of the patients were on vasopressor and ionotropic drugs. Intuitively, sympathetic activation is associated with increases in both PPI and blood pressure. The blood pressure management might lead to similar blood pressure changes between the two treatment groups. Although repetitive taVNS increases HRV days after initiation of the treatment, this effect is not seen acutely. Also, while repetitive taVNS was associated with a reduced PPI and no change in heart rate and blood pressure, there were small acute increases in PPI, heart rate, and blood pressure. All patients who were capable of verbal communication were asked if they felt any prickling or pain during all sessions. We confirmed that the current taVNS protocol is below the perception threshold for all trialed patients. Altogether, successful activation of the afferent vagal pathway by taVNS increased arousal, resulting in increased heart rate (*Naredi et al., 2000*; *Skora et al., 2024Sharon et al., 2021*; *Skora et al., 2024*; *Tan et al., 2024*). These speculative mechanisms warrant further validation through animal or pharmacological studies directly investigating the effects of taVNS on autonomic function and vascular tone.

## Considerations for applying taVNS on SAH patients

Blood pressure management and cardiac function monitoring are crucial in patients following SAH (*Minhas et al., 2022*). This study shows that blood pressure, QT interval, and heart rate over days were not significantly different between taVNS and sham treatment groups. This suggests that adding taVNS in treatments for SAH patients is unlikely to cause adverse blood pressure alterations or cardiac complications. Our findings suggest that repetitive taVNS could enhance parasympathetic tone following SAH. This effect could lead to favorable clinical outcomes as lower HRV was found to be associated with neurocardiogenic injury (*Bai et al., 2023*; *Megjhani et al., 2020*). Given the negative association between pro-inflammatory markers and HRV, our finding that HRV was higher in the taVNS treatment group aligns with the findings of primary outcomes of this clinical trial, which showed that taVNS treatment reduced pro-inflammatory cytokines, including TNF-α and interleukin-6 (*Huguenard et al., 2024b*; *Williams et al., 2019*). The consistency between these findings strengthens the evidence supporting the anti-inflammatory effects of taVNS. In addition, the sympathetic predominance following SAH is implicated in an increased risk of delayed cerebral vasospasm, which is most commonly detected 5–7 days after SAH (*Budohoski et al., 2013*). Given that taVNS treatment mitigated the sympathetic overactivation before the typical onset of cerebral vasospasm, it could potentially reduce the severity of this complication. Additionally, reduced PPI was associated with increased respiration rate only for the taVNS treatment group, suggesting that the autonomic system self-regulates to maintain cardiovascular homeostasis. Thus, it is important to consider autonomic system self-regulation when studying the therapeutic effects of taVNS (*Bjerkne Wenneberg et al., 2020*). Also, while acute cardiovascular changes were noted after taVNS, these changes were within normal daily variations in this study, making them unlikely to pose a risk to the patient. That said, the observed acute increases in PPI following taVNS necessitate caution when considering taVNS treatment for patients to whom peripheral vasodilatation is not desired.

As we pioneer the application of taVNS as an immunomodulation technique in SAH patients, we adopt parameters (20 Hz, 0.4 mA) reported in similar studies (*Jelinek et al., 2024*). The current study provides a basis for future preclinical and clinical studies of taVNS in this patient population. To build on our findings, a systematic evaluation of the relationship between parameters such as frequency, intensity, and duration and taVNS's effects on the immune system and cardiovascular function is necessary to establish taVNS as an effective therapeutic option for SAH patients (*Dusi et al., 2022*).

## Limitations and outlook

While this study supports the safety of taVNS treatment in SAH patients, we should be cautious when generalizing these findings to broader clinical populations. The current study did not explore the effects of taVNS on less commonly used cardiovascular metrics, such as QTc dispersion. Our study considers each day as an independent sample for the following considerations: (1) heart rate and HRV metrics exhibited great daily variations. Their value on 1 day was not predictive of the metrics on another day, which could be due to medications, interventions, or individualized SAH recovery process during the patient's stay in the ICU. (2) SAH patients in the ICU often experience daily changes in clinical status, including fluctuations in ICP, blood pressure, neurological status, and other vital signs. (3) Day-to-day cardiovascular function changes varied as the patient recovered or encountered setbacks. To conclusively establish that there is no significant cardiovascular effect of repetitive taVNS on any given day following SAH, we would need to perform statistical tests between treatment groups for each day. In this context, 64 subjects per treatment group are required to achieve 80% power, assuming a medium effect size (Cohen's $d$ = 0.5)and 0.05 type I error probability (two-sample $t$-test).

Mild cardiac abnormalities are common in SAH patients (*Norberg et al., 2018*), complicating the precise calculation of cardiovascular metrics from ECG signals and the interpretation of the results. Systematic verification of methods for calculating cardiovascular metrics to ensure their applicability in SAH patients is crucial. We noticed a high variance of change in heart rate for Days 5–7, 8–10, and 11–13 for both treatment groups (*Figure 2D*). This may be due to the small sample size in the later days, given that the mean duration of hospitalization for the 24 subjects included in this study was 11.3 days with a standard deviation of 6.4. Differences in medical history and clinical outcomes during hospitalization may also explain the variance of change in heart rate for the later days. For example. heart rate was lower in patients with improved mRS scores (*Figure 3—figure supplement 4B*). Understanding the association between cardiovascular metrics and clinical assessments, such as vasospasm and inflammation, could help decide whether future taVNS trials should control for these factors when evaluating the effects of taVNS on cardiovascular function. Additional care should be paid when interpreting the results of blood pressure, as hypertension was intentionally induced for some patients being treated for vasospasm. Patient medical histories are summarized in *Table 1*.

Kulkarni et al. showed that the response to low-level tragus stimulation varied among patients with atrial fibrillation (*Kulkarni et al., 2021*). Similarly, in our study, not all patients in the taVNS treatment group showed a reduction in mRS scores (improved degree of disability or dependence). This differential response may be inherent to taVNS and potentially influenced by factors such as anatomical variations in the distribution of the vagus nerve at the outer ear. These findings underscore the importance of using acute biomarkers to guide patient selection and optimize stimulation parameters. Furthermore, we found that increased heart rate was a potential acute biomarker for identifying SAH patients who are most likely to respond favorably to taVNS treatment. Translating this finding into clinical practice will require further research to elucidate the mechanisms by which an acute increase in heart rate may predict the outcomes of patients receiving taVNS, including its relationship with neurological evaluations, vasospasm, echocardiography, and inflammatory markers.

## Conclusions

Utilizing taVNS as a neuromodulation technique in SAH patients is safe without inducing bradycardia or QT prolongation. Repetitive taVNS treatment increased parasympathetic activity. Acute taVNS elevated heart rate, which might be an acute biomarker to identify SAH patients who are likely to respond favorably to taVNS treatment.

## Methods

### Study participants

Participants in this study were recruited from adult patients who were admitted to the ICU at Barnes Jewish Hospital, St. Louis, MO, following an acute, spontaneous, aneurysmal SAH. Inclusion criteria were: (1) Patients with SAH confirmed by CT scan; (2) Age >18; (3) Patients or their legally authorized representative are able to give consent. Exclusion criteria were: (1) Age <18; (2) Use of immuno-suppressive medications; (3) Receiving ongoing cancer therapy; (4) Implanted electrical device; (5) Sustained bradycardia on admission with a heart rate <50 beats per minute for >5 min; (6) Considered moribund/at risk of imminent death. Participants were randomized to receive either the taVNS ($N$ = 11) or Sham ($N$ = 13) treatment. Patients were enrolled prior to randomization by a member of the research team who went through the informed consent process with the patient or their legally authorized representative. Treatment group assignment was via a computer-generated randomization sequence, with the next number obscured until patient enrollment. Research team members who applied the ear clips and set stimulation parameters were not blinded to the treatment. The participants, the medical team who dictated all management decisions for the patient's SAH, and the outcomes assessors who assigned mRS at admission and discharge were blinded to the treatment. The structure of this study is shown in *Figure 1B*. This study was approved by the Washington University School of Medicine Review Board and was conducted in accordance with institutional and national ethics guidelines and the Declaration of Helsinki (Clinical trial number: NCT04557618).

### taVNS protocol

Following randomization, enrolled patients underwent 20 min of either taVNS or sham stimulation twice daily during their stay in the ICU. During treatment periods, a portable TENS device (TENS 7000 Digital TENS Unit, Compass Health Brands, OH, USA) was connected to the patient's left ear using two ear clips (*Figure 1C, D*). For taVNS treatments, these ear clips were placed along the concha of the ear, while for sham treatments, the clips were placed along the earlobe to avoid stimulation of the auricular vagus nerve from tactile pressure (*Figure 1C*). For the taVNS group, stimulation parameters were selected based on values reported in prior studies that sought to maximize vagus somatosensory evoked potentials while avoiding the perception of pain: 20 Hz frequency, 250 μs pulse width, and 0.4 mA intensity (*de Gurtubay et al., 2021*). The stimulation was not perceptible for the patients. No electrical current was delivered during sham treatments. For both groups, the TENS device was connected to the patient and a bedside recording computer. The computer recorded continuous ECG and vital signs, including blood pressure, temperature, respiration rate, PPI, ICP, and arterial blood pressure. The collection of ICP and arterial blood pressure data varied, being dependent on the treatment protocol assigned by the clinical team, and thus was not uniformly available for all patients throughout the study. Please see *Huguenard et al., 2024a* for a detailed protocol of this study.

### Data processing

A 3-lead system was used for ECG. ECG signals, sampled at 500 Hz, and other vital signs, such as blood pressure, sampled at 1 Hz, were recorded from the Intellivue patient monitor (Philips, Netherlands) using vitalDB software (*Lee and Jung, 2018*).

To calculate cardiac metrics, we first applied a 0.5-Hz fifth-order high-pass Butterworth filter and a 60-Hz powerline filter on ECG data to reduce artifacts (*Makowski et al., 2021*). We detected QRS complexes based on the steepness of the absolute gradient of the ECG signal using the Neurokit2 software package (*Makowski et al., 2021*). R-peaks were detected as local maxima in the QRS complexes. P waves, T waves, and QRS complexes were delineated based on the wavelet transform of the ECG signals proposed by *Figure 2A–C*; *Martínez et al., 2004*. This algorithm identifies the QRS complex by searching for modulus maxima, which are peaks in the wavelet transform coefficients that exceed specific thresholds. The onset of the QRS complex is determined as the beginning of the first modulus maximum before the modulus maximum pair created by the R wave. To identify the T wave, the algorithm searches for local maxima in the absolute wavelet transform in a search window defined relative to the QRS complex. Thresholding is used to identify the offset of the T wave. RR intervals were preprocessed to exclude outliers, defined as RR intervals greater than 2 s or less than 300 ms. RR intervals with >20% relative difference to the previous interval were considered ectopic

beats and excluded from analyses. After preprocessing, RR intervals were used to calculate heart rate, HRV, and corrected QT (QTc) based on Bazett's formula: $QTc = \frac{QT}{\sqrt{RR}}$ (**Bazett, 1920**). The corrected QT interval (QTc) estimates the QT interval at a standard heart rate of 60 bpm. HRV measures included the root mean square of successive difference of normal RR intervals (RMSSD), indicating parasympathetic activity, and the standard deviation of normal RR intervals (SDNN), a clinical measure of cardiac risk (**Kleiger et al., 2005**; **Shaffer and Ginsberg, 2017**). HRV calculations are detailed in Appendix 1.

To investigate the effect of repetitive taVNS on cardiovascular function, we compared HRV, heart rate, corrected QT intervals, blood pressure, and ICP calculated over 24 hr between patients receiving taVNS and sham treatment. In addition, we compared the mean PPI and respiration rate over 24 hr between treatment groups to determine the effects of repetitive taVNS on the autonomic system. Data collection commenced on the first day of each patient's ICU admission. The average duration of continuous data recording was 11.1 days, with a standard deviation of 6.8 days. To analyze the effects of taVNS treatment more granularly, we segmented the changes in these metrics from the initial day at 3-day intervals, facilitating comparison between the taVNS and sham treatment groups over the course of their ICU stay.

To study the effects of acute treatment over time, we focused on blood pressure, HRV, heart rate, and corrected QT intervals 20 min before treatment (pre-treatment), during the 20 min treatment (during-treatment), and 20 min after treatment (post-treatment). The treatment event signals were rectified and binarized based on their half-maximum value to identify the treatment onset and offset (**Figure 2A**). We calculated metrics using 6 min sliding windows over ECG data starting from treatment onset/offset and moving bi-directionally with a 3-min step. To correct daily and between-subject variation, we applied the same sliding window strategy to calculate the mean and standard deviation of these cardiac metrics for each patient each day as a reference. Subsequently, HRV, heart rate, and corrected QT interval around treatment onset/offset were normalized based on the reference. In addition, we calculated the difference in blood pressure, HRV, and heart rate, and corrected QT intervals between during- and pre-treatment, as well as the difference between post- and pre-treatment for each patient and for each treatment. To study the effects of acute taVNS, we compared the two differences between the treatment groups.

## Factor analysis

We performed an exploratory factor analysis to identify the factors underlying autonomic system activity. Besides RMSSD and SDNN, variables derived from preprocessed RR intervals and used to perform factor analysis included the percentage of successive normal-to-normal (NN) Intervals that differ by more than 50 ms (pNNI_50), total power (below 0.4 Hz), normalized high-frequency power (0.15–0.4 Hz), cardiac vagal index, and CSI. The total power is thought to represent the overall HRV, while normalized high-frequency power primarily reflects parasympathetic activity (**Kleiger et al., 2005**). These variables were normalized using a *z*-score method based on individual daily means and standard deviations before factor analysis. Factor analysis was performed using the factor_analyzer Python package. The number of factors was set to 2 based on the Scree plot. The factor loadings were calculated using the Minimum Residual Method. After factor extraction, a Varimax rotation was applied for better interpretability so that each factor had high loadings for a smaller number of variables and low loadings for the remaining variables.

## Statistical analyses

To investigate the effect of taVNS at the phase of early brain injury and later phases, we grouped the change of HRV, heart rate, and corrected QT interval from the first hospitalized day in bins of 3 days. The change in blood pressure, ICP, respiration rate, and PPI from the first hospitalization day were also compared between treatment groups. We used *t*-tests for comparisons between treatment groups when the data were normally distributed, as determined by the Shapiro–Wilk test. We employed Mann–Whitney *U* tests for non-normally distributed data. We used Wilcoxon signed-rank tests to compare the difference in heart rate between post- and during-treatment against 0. To control the familywise error rate, we applied Bonferroni correction. Specifically, when investigating the cardiac effects of taVNS, we compared six metrics between treatment groups, including heart rate, corrected QT interval, RMSSD, SDNN, and two factors representing HRV. Consequently, the p-values were corrected by a factor of six. In this study, we reported the statistical power achieved

for tests that yielded non-significant results. The achieved power is calculated based on a two-sample *t*-test assuming a medium effect size (Cohen's *d* of 0.5) and a type I error probability (a) of 0.05. We used two one-sided tests to confirm that taVNS did not induce long-term changes in heart rate, corrected QT interval, or blood pressure, with equivalency test margins set to 5 bpm for heart rate, 50 ms for QT interval, and 2 mmHg for blood pressure. A summary of statistical tests is provided in *Appendix 2—table 1*.

## Acknowledgements

The authors acknowledge physicians/nurses for helping administer treatment. The authors thank Dr. Paul Cassidy for his contributions to the scientific editing of this manuscript, supported by the Institute of Clinical and Translational Sciences grant UL1TR002345 from the National Center for Advancing Translational Sciences (NCATS).

## Additional information

### Competing interests

Anna L Huguenard: Has stock ownership in Aurenar. Eric C Leuthardt: Has stock ownership in Neurolutions, Face to Face Biometrics, Caeli Vascular, Acera, Sora Neuroscience, Inner Cosmos, Kinetrix, NeuroDev, Inflexion Vascular, Aurenar, Cordance Medical, Silent Surgical, and Petal Surgical; consultant for E15, Neurolutions, Inc, Petal Surgical; Washington University owns equity in Neurolutions. The other authors declare that no competing interests exist.

### Funding

| Funder | Grant reference number | Author |
| --- | --- | --- |
| American Association of Neurological Surgeons | | Anna L Huguenard |
| Aneurysm and AVM Foundation | | Anna L Huguenard |
| National Institutes of Health | R01-EB026439 | Peter Brunner |
| National Institutes of Health | P41-EB018783 | Peter Brunner |
| National Institutes of Health | U24-NS109103 | Peter Brunner |
| National Institutes of Health | R21-NS128307 | Eric C Leuthardt |
| McDonnell Center for Systems Neuroscience | | Peter Brunner / Eric C Leuthardt |
| Fondazione Neurone | | Peter Brunner |

The funders had no role in study design, data collection and interpretation, or the decision to submit the work for publication.

### Author contributions

Gansheng Tan, Conceptualization, Data curation, Software, Formal analysis, Validation, Investigation, Visualization, Methodology, Writing – original draft, Writing – review and editing; Anna L Huguenard, Conceptualization, Data curation, Formal analysis, Funding acquisition, Investigation, Project administration, Writing – review and editing; Kara M Donovan, Methodology, Writing – review and editing; Phillip Demarest, Visualization, Methodology, Writing – review and editing; Xiaoxuan Liu, Ziwei Li, Validation, Writing – review and editing; Markus Adamek, Data curation, Software; Kory Lavine, Validation; Ananth K Vellimana, Terrance T Kummer, Joshua W Osbun, Data curation, Validation, Project administration; Gregory J Zipfel, Funding acquisition, Validation, Project administration; Peter Brunner, Resources, Software, Supervision, Funding acquisition, Validation, Project administration;

Eric C Leuthardt, Conceptualization, Resources, Supervision, Funding acquisition, Validation, Investigation, Methodology, Project administration, Writing – review and editing

**Author ORCIDs**
Gansheng Tan ⓘ https://orcid.org/0000-0001-8785-9499
Markus Adamek ⓘ https://orcid.org/0000-0001-8519-9212
Terrance T Kummer ⓘ https://orcid.org/0000-0001-8938-8280
Peter Brunner ⓘ https://orcid.org/0000-0002-2588-2754
Eric C Leuthardt ⓘ https://orcid.org/0000-0003-4154-3135

**Ethics**
Clinical trial registration NCT04557618.
This study was approved by the Washington University School of Medicine Review Board and was conducted in accordance with institutional and national ethics guidelines and the Declaration of Helsinki (Clinical trial number: NCT04557618). Participants or their legally authorized representatives provided written informed consent prior to enrollment in the study.

Reviewer #2 (Public review): https://doi.org/10.7554/eLife.100088.3.sa1
Reviewer #3 (Public review): https://doi.org/10.7554/eLife.100088.3.sa2
Author response https://doi.org/10.7554/eLife.100088.3.sa3

# Additional files

**Supplementary files**
MDAR checklist

**Data availability**
All relevant data has been made publicly available on Dryad at https://doi.org/10.5061/dryad.rfj6q57kw. Codes for generating the figures are available at https://github.com/GanshengT/taVNS_SAH (copy archived at *Tan, 2024*).

The following dataset was generated:

| Author(s) | Year | Dataset title | Dataset URL | Database and Identifier |
|---|---|---|---|---|
| Tan G, Huguenard AL, Brunner P, Zipfel GJ, Leuthardt EC | 2024 | Electrocardiography and vital signs recorded in patients with subarachnoid hemorrhage (NCT04557618) | https://doi.org/10.5061/dryad.rfj6q57kw | Dryad Digital Repository, 10.5061/dryad.rfj6q57kw |

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

## Appendix 1

### Calculation of cardiovascular metrics

1. Heart rate (HR) over a period is the mean of $\frac{60}{NN\,interval\,in\,seconds}$ over this period.

2. The percentage of prolonged QT intervals is defined as the number of intervals with a corrected QT interval (QTc) ≥500 ms, divided by the total number of QT intervals.

3. Root mean square of successive difference of normal RR intervals (RMSSD) for a given period:

$RMSSD = \sqrt{\frac{1}{N-1} \sum_{i}^{N-1} \left(NN_{i+1} - NN_i\right)^2}$, where $N$ is the number of NN intervals within this period.

4. Standard deviation of normal RR intervals (SDNN) for a given period:

$SDNN = \sqrt{\frac{1}{N} \sum_{i}^{N} \left(NN_i - \overline{NN}\right)^2}$, where $\overline{NN}$ represents the mean of NN intervals within this period.

5. To calculate total power and normalized high-frequency power, power spectral density is calculated from NN intervals with the help of Python package *hrv-analysis*. Linear interpolation is used to make NN intervals evenly spaced, assuming a sampling frequency of 4 Hz. Welch's method is used to calculate power spectral density $P\left(f\right)$.

$$Total\,power = \int_{0}^{0.4} P\left(f\right)\,df$$

$$High-frequency\,power = \int_{0.15}^{0.4} P\left(f\right)\,df$$

$$Normalized\,high-frequency\,power = \frac{High-frequency\,power}{Total\,power}$$

6. Cardiac vagal index (CVI) and cardiac sympathetic index (CSI):

The width (SD1) and length (SD2) of the Poincaré plot, a graphical representation of the correlation between consecutive NN intervals, are used to calculate CVI and CSI.

$$CSI = \frac{SD1}{SD2}$$

$$CVI = log_{10}\left(SD1 * SD2\right)$$

# Appendix 2

**Appendix 2—table 1.** Summary of statistical tests.
Bolded values were used to make inferences in this paper.

| Test name | Variable | Distribution A | Distribution B | N(A) | N(B) | p | Statistics | Effect size |
|---|---|---|---|---|---|---|---|---|
| T | RMSSD change | taVNS | Sham | 94 | 95 | **0.004** | t = 2.91 | 0.42 |
| T | SDNN change | taVNS | Sham | 94 | 95 | 0.48 | t = 0.71 | 0.10 |
| Mann–Whitney U | HR change | taVNS | Sham | 94 | 95 | 0.69 | U(A) = 4317 | −0.01 |
| Equivalence test (two one-sided t-tests) | HR change | taVNS | Sham | 94 | 95 | **0.006 (lower), 0.004 (upper)** | 2.53 (lower) −2.72 (upper) | −0.01 |
| Mann–Whitney U | QTc change | taVNS | Sham | 94 | 95 | **<0.01** | U(A) = 3539 | **−0.57** |
| Equivalence test (two one-sided t-tests) | QTc change margin: 30 ms | taVNS | Sham | 94 | 95 | 0.50 (lower), **1.45*10$^{-13}$ (upper)** | • −0.004 (lower) • −7.86 (upper) | −0.57 |
| Mann–Whitney U | Percentage of prolonged QT | taVNS | Sham | 94 | 95 | **<0.01** | U(A) = 2885 | **−0.72** |
| Mann–Whitney U | Overall HRV change | taVNS | Sham | 94 | 95 | **0.04** | U(A) = 5237 | **0.37** |
| Mann–Whitney U | Parasympathetic activity change | taVNS | Sham | 94 | 95 | **0.04** | U(A) = 5238 | 0.29 |
| Mann–Whitney U | ICP change | taVNS | Sham | 66 | 52 | 0.61 | U(A) = 1972 | 0.25 |
| Equivalence test (two one-sided t-tests) | ICP change margin: 2 mmHg | taVNS | Sham | 66 | 52 | **3.66*10$^{-13}$ (lower), 3.33*10$^{-10}$ (upper)** | 8.07 (lower) −6.73 (upper) | 0.12 |
| Mann–Whitney U | BP change | taVNS | Sham | 66 | 81 | 0.73 | U(A) = 2842 | −0.11 |
| Equivalence test (two one-sided t-tests) | BP change margin: 2 mmHg | taVNS | Sham | 66 | 81 | 0.07 (lower), 0.002 (upper) | 1.51 (lower) −3.00 (upper) | −0.12 |
| Mann–Whitney U | SBP change | taVNS | Sham | 66 | 81 | 0.90 | U(A) = 2719 | −0.07 |
| Mann–Whitney U | DBP change | taVNS | Sham | 66 | 81 | 0.73 | U(A) = 2846 | −0.17 |
| Mann–Whitney U | ABP change | taVNS | Sham | 28 | 24 | 0.46 | U(A) = 295 | −0.48 |
| Mann–Whitney U | PPI change | taVNS | Sham | 83 | 95 | **0.002** | U(A) = 2877 | −0.49 |
| Mann–Whitney U | RR change | taVNS | Sham | 94 | 95 | **0.004** | U(A) = 5530 | 0.37 |
| Mann–Whitney U | HR difference (post–pre) | taVNS | Sham | 188 | 199 | 0.28 | U(A) = 17,527 | 0.10 |
| Wilcoxon signed rank | HR difference (post–pre) | taVNS | | 188 | | **0.02** | 7525 | 0.11 |
| Mann–Whitney U | QTc difference (post–pre) | taVNS | Sham | 188 | 198 | 0.86 | U(A) = 18,412 | 0.02 |
| Mann–Whitney U | RMSSD difference (post–pre) | taVNS | Sham | 188 | 199 | 0.31 | U(A) = 17,581 | 0.14 |
| Mann–Whitney U | SDNN difference (post–pre) | taVNS | Sham | 188 | 199 | 0.48 | U(A) = 17,923 | 0.03 |
| Mann–Whitney U | HR difference (post–pre) | mRS change <−1 | mRS change ≥−1 | 53 | 122 | **0.01** | U(A) = 4019 | 0.38 |
| Mann–Whitney U | BP difference (post–pre) | taVNS | Sham | 159 | 180 | **0.03** | U(B) = 12,400 | 0.21 |

*Appendix 2—table 1 Continued on next page*

*Appendix 2—table 1 Continued*

| Test name | Variable | Distribution A | Distribution B | N(A) | N(B) | p | Statistics | Effect size |
|---|---|---|---|---|---|---|---|---|
| Mann–Whitney U | ICP difference (post–pre) | taVNS | Sham | 146 | 114 | 0.82 | U(B) = 12,400 | 0.09 |
| Mann–Whitney U | PPI difference (post–pre) | taVNS | Sham | 186 | 227 | 0.002 | U(B) = 17,386 | 0.19 |
| Mann–Whitney U | RR difference (post–pre) | taVNS | Sham | 214 | 224 | 0.10 | U(B) = 21,806 | 0.11 |

# Appendix 3

**Appendix 3—table 1.** Clinical characteristics and cardiovascular metrics of the two arms. Values in this table represent the median.

|  | taVNS | Sham |
|---|---|---|
| Age | 67 | 53 |
| % of female | 63.6% | 84.6% |
| % with known hypertension | 90.9% | 46.2% |
| % with known diabetes mellitus | 18.2% | 7.7% |
| % with arrhythmia PTA | 9.1% | 7.7% |
| % with coronary artery disease PTA | 0% | 15.4% |
| % on beta blockers PTA | 27.3% | 38.5% |
| % on calcium channel blockers PTA | 27.3% | 7.7% |
| % on angiotensin-converting enzyme inhibitors PTA | 27.3% | 15.4 |
| Heart rate (bpm) | 86.0 | 78.8 |
| Change from pre- to post-treatment | 1.1 | 0.3 |
| Change from pre- to during-treatment | 0.4 | –0.2 |
| QTc (ms) | 509.9 | 483.6 |
| Change from pre- to post-treatment | –0.4 | –0.2 |
| Change from pre- to during-treatment | 1.3 | 1.0 |
| SDNN (ms) | 72.7 | 77.1 |
| Change from pre- to post-treatment | –0.4 | –1.7 |
| Change from pre- to during-treatment | –1.0 | –1.4 |
| RMSSD (ms) | 24.8 | 24.5 |
| Change from pre- to post-treatment | –0.1 | –0.4 |
| Change from pre- to during-treatment | –0.3 | –0.8 |
| Relative power of high-frequency band (%) | 30.4 | 30.7 |
| Change from pre- to post-treatment | –0.4 | –0.7 |
| Change from pre- to during-treatment | –0.5 | –0.3 |
| Blood pressure (mmHg) | 92.4 | 95.3 |
| Change from pre- to post-treatment | 1.8 | 0.0 |
| Change from pre- to during-treatment | 0.0 | 0.0 |
| PPI (a.u.) | 1.6 | 2.3 |
| Change from pre- to post-treatment | 0.0 | –0.1 |
| Change from pre- to during-treatment | 0.0 | –0.1 |
| ICP (mmHg) | 6 | 6.3 |
| Change from pre- to post-treatment | 0.0 | 0.0 |
| Change from pre- to during-treatment | 0.0 | 0.0 |
| Respiration rate (bpm) | 20.1 | 18.3 |
| Change from pre- to post-treatment | 0.1 | 0.1 |
| Change from pre- to during-treatment | 0.0 | 0.0 |

QTc: corrected QT interval. PPI: peripheral perfusion index. ICP: intracranial pressure. PTA: prior to admission.

# Appendix 4

**Appendix 4—table 1.** Effect of taVNS on cardiac metrics adjusted for age.
The regression formula is: cardiac metric ~ Treatment + Age.

|  | Coefficient | p-value | 95% CI |
|---|---|---|---|
| *Heart rate change from Day 1 (F = 9.49, p < 0.01 for the overall model)* | | | |
| taVNS treatment | 1.87 | 0.21 | [–1.05, 4.80] |
| Age | –0.26 | <0.01 | [–0.37, –0.14] |
| *QTc change from Day 1 (F = 14.23, p < 0.01 for the overall model)* | | | |
| taVNS treatment | –30.79 | <0.01 | [–42.16, –19.43] |
| Age | 0.19 | 0.40 | [–0.26, 0.65] |
| *SDNN change from Day 1 (F = 29.40, p < 0.01 for the overall model)* | | | |
| taVNS treatment | 15.43 | <0.01 | [5.14, 25.72] |
| Age | –1.56 | <0.01 | [–1.97, –1.15] |
| *RMSSD change from Day 1 (F = 25.09, p < 0.01 for the overall model)* | | | |
| taVNS treatment | 16.90 | <0.01 | [10.02, 23.78] |
| Age | –0.83 | <0.01 | [–1.10, –0.56] |

