## [Editor Report · eLife Assessment]

The authors provide a **solid** set of data supporting the safety of transcutaneous auricular vagal nerve stimulation on cardiovascular parameters in the acute setting of critically ill patients presenting with subarachnoid hemorrhage. This **important** study also suggests a promising effect on autonomic balance.

---

## [Referee Report · Reviewer #2 (Public review)]

Summary:

This study investigated the effects of transcutaneous auricular vagus nerve stimulation (taVNS) on cardiovascular dynamics in subarachnoid hemorrhage (SAH) patients. The researchers conducted a randomized clinical trial with 24 SAH patients, comparing taVNS treatment to a Sham treatment group (20 minutes per day twice a day during the ICU stay). They monitored electrocardiogram (ECG) readings and vital signs to assess acute as well as middle -term changes in heart rate, heart rate variability, QT interval, and blood pressure between the two groups. The results showed that repetitive taVNS did not significantly alter heart rate, corrected QT interval, blood pressure, or intracranial pressure. However, it increased overall heart rate variability and parasympathetic activity after 5-10 days of treatment compared to the sham treatment. Acute taVNS led to an increase in heart rate, blood pressure, and peripheral perfusion index without affecting corrected QT interval, intracranial pressure, or heart rate variability. The acute post-treatment elevation in heart rate was more pronounced in patients who showed clinical improvement. In conclusion, the study found that taVNS treatment did not cause adverse cardiovascular effects, suggesting it as a safe immunomodulatory treatment for SAH patients. The mild acute increase in heart rate post-treatment could potentially serve as a biomarker for identifying SAH patients who may benefit more from taVNS therapy.

Strengths:

The paper is overall well written, and the topic is of great interest. The methods are solid and the presented data are convincing.

Comments on revisions:

The main previous weaknesses of the paper have now been fixed.

---

## [Referee Report · Reviewer #3 (Public review)]

Summary:

The authors characterized the cardiovascular effects of acute and repetitive taVNS as an index of safety and concluded that taVNS treatment does not induce adverse cardiovascular effects such as bradycardia or QT prolongation.

Strengths:

This study contributes important information about the clinical utility of taVNS as a safe immunomodulatory treatment approach for SAH patients.

Comments on revised version:

A number of limitations were identified previously: https://elifesciences.org/reviewed-preprints/100088/reviews#peer-review-2. These major concerns were largely addressed by the authors.

---

## [Author Response]

The following is the authors’ response to the original reviews.

We are thankful to the reviewers and the editor for their detailed feedback, insightful suggestions, and thoughtful assessment of our work. The revised manuscript has taken into account all the comments of the three reviewers. We have also undertaken additional analyses and added materials in response to reviewer suggestions. In brief:

(1) We have conducted a more in-depth analysis of frequency domain HRV metrics to better depict the change of autonomic tone.

(2) We have revised the manuscript to provide justifications for the chosen taVNS protocol and to clearly articulate the objectives of the current study.

(3) In response to comments from reviewer #2, we have included two new tables that present the absolute changes in cardiovascular metrics, clinical characteristics for the two trial arms, and effects of taVNS adjusted for age.

Other significant amendments include:

(1) An expanded discussion linking our findings to the existing literature on the effects of taVNS on cardiovascular function, biomarkers for taVNS response, the safety of taVNS, and the dose-response relationship of taVNS.

(2) Revision to the Method section to provide details of QT interval calculation.

**Reviewer #1 (Public Review):**
The authors report the results of a randomized clinical trial of taVNS as a neuromodulation technique in SAH patients. They found that taVNS appears to be safe without inducing bradycardia or QT prolongation. taVNS also increased parasympathetic activity, as assessed by heart rate variability measures. Acute elevation in heart rate might be a biomarker to identify SAH patients who are likely to respond favorably to taVNS treatment. The latter is very important in light of the need for acute biomarkers of response to neuromodulation treatments.Comments:(1) Frequency domain heart rate variability measures should be analyzed and reported. Given the short duration of the ECG recording, the frequency domain may more accurately reflect autonomic tone.

We sincerely appreciate this encouraging summary of our paper. We have analyzed and reported frequency-domain heart rate variability measures, including the relative power of the high-frequency band (0.15–0.4 Hz) and the relative power of the low-frequency band (0.04 – 0.15). We showed the distribution of the two frequency-domain HRV measures in supplementary Figure 2C-D. For 24-hour ECG recording, we found that the change in the relative high-frequency power from Day 1 was not significantly different between the treatment groups. As both high-frequency band and low-frequency band power are relative to the total power, the comparison of the relative power of the low-frequency band between groups would be the opposite of the relative power of the high-frequency band. As both time-domain and frequency-domain HRV measures can reflect the autonomic tone, we performed factor analysis to identify the parasympathetic activity component (Figure 2D). Comparing the change in parasympathetic activity component and relative high-frequency power, we observed similarities and discrepancies. Specifically, both the change in parasympathetic activity component and the change in relative high-frequency power were higher in the taVNS group at the early phase (Day 2 - 4).

We also observed higher high-frequency power in the Sham group at the later phase. If the factor analysis successfully isolates the parasympathetic activity, there should be other factors than the parasympathetic activity affecting the relative power of the high-frequency band. One such factor is the respiration rate. The high-frequency range is between 0.15 to 0.4 Hz, corresponding to respiration's frequency range of approximately 9 to 24 breaths per minute. If the respiration rate increases and exceeds 24 breaths per minute, the respiratory-driven HRV might occur at a frequency higher than the typical high-frequency band. Given that the respiration rate was higher in the taVNS treatment group, a compensatory mechanism to ensure oxygen delivery (Figure 4E), we hypothesized that observed lower high-frequency power in the taVNS treatment group compared to sham at later phases is a result of increased respiration rate in the taVNS treatment group. Indeed, we found the normalized high-frequency power is higher when RR is less than 25 bpm compared to when RR > 25 bpm (Cohen’s d = 0.85, Supplementary Figure 3A). Moreover, an increase in RR in the taVNS treatment group is associated with a decrease in high-frequency power (Supplementary Figure 3B). These control analyses underscored the necessity of performing factor analysis to robustly measure parasympathetic activities and confirm that taVNS treatment mitigated the sympathetic overactivation during the early phase.

We have now discussed the results of frequency-domain HRV measures in the Discussion section: taVNS and autonomic system (p23): “A key metric that reflects this restored sympathovagal balance is the increase in heart rate variability (Figure 3F). Specifically, the factor analysis showed that the parasympathetic activity was significantly higher in the taVNS treatment group. This difference was most pronounced during the early phase, particularly between Days 2 and 4 following SAH. In addition to analyzing the correlation between the parasympathetic activity factor and established HRV measures that reflect parasympathetic activity such as RMSSD and pNNI_50 (Figure 3C), we also examined changes in a frequency-domain HRV measure—the relative power of the high-frequency band (0.15–0.4 Hz)—to validate the accuracy of the factor analysis. the relative power of the high-frequency band is widely used to indicate respiratory sinus arrhythmia, a process primarily driven by the parasympathetic nervous system (Supplementary Figure 2). We found that both the change in parasympathetic activity factor and relative high-frequency power were higher in the taVNS group at the early phase (Day 2 - 4). Conversely, we observed higher high-frequency power in the Sham group during the later phase. If the factor analysis successfully isolates the parasympathetic activity, there should be other factors than the parasympathetic activity affecting the relative power of the high-frequency band. One such factor is the respiration rate. The high-frequency range is between 0.15 to 0.4 Hz, corresponding to respiration's frequency range of approximately 9 to 24 breaths per minute. If the respiration rate increases and exceeds 24 breaths per minute, the respiratory-driven HRV might occur at a frequency higher than the typical high-frequency band. Given that the respiration rate was higher in the taVNS treatment group, a compensatory mechanism to ensure oxygen delivery (Figure 4E), we hypothesized that observed lower high-frequency power in the taVNS treatment group compared to sham at later phases is a result of increased respiration rate in the taVNS treatment group. Indeed, we found the normalized high-frequency power is higher when RR is less than 25 bpm compared to when RR > 25 bpm (Cohen’s d = 0.85, Supplementary Figure 3A). Moreover, an increase in RR in the taVNS treatment group is associated with a decrease in high-frequency power (Supplementary Figure 3B). These control analyses underscored the necessity of performing factor analysis to robustly measure parasympathetic activities and confirm that taVNS treatment mitigated the sympathetic overactivation during the early phase.”

We have also reported the changes in the relative power of the high-frequency band between the two treatment groups in Supplementary Figure 6. We did not find a significant change in relative high-frequency band power between the treatment groups (Treatment – pre-treatment difference: p = 0.74, Cohen’s d = -0.08, N(Sham) = 199, N(taVNS) = 188, Mann-Whitney U test). We reported these results in the Results section: Acute effects of taVNS on cardiovascular function (p18): “There were no significant differences in changes in corrected QT interval or heart rate variability, as measured by RMSSD, SDNN, and relative power of high-frequency band between treatment groups (Figure 5D and E and Supplementary Figure 6).”

How was the "dose" chosen (20 minutes twice daily)?

The choice of a 20-minute taVNS session twice daily was informed by findings from Addorisio et al. (2019), where the authors administered 5-minute taVNS twice daily to patients with rheumatoid arthritis for two days. They found that the circulating c-reactive protein (CRP) levels significantly reduced after two days of treatment but returned to baseline at the second clinical assessment by day 7. Given the high inflammatory state associated with subarachnoid hemorrhage (SAH) and our intention to maintain a steady reduction in inflammation, we extended the duration of taVNS to 20 minutes per session. We have clarified this stimulation schedule's rationale in the Results section (p5-6): “This treatment schedule was informed by findings from Addorisio et al., where a 5-minute taVNS protocol was administered twice daily to patients with rheumatoid arthritis for two days.29 Their study found that circulating c-reactive protein (CRP) levels significantly reduced after 2 days of treatment but returned to baseline at the second clinical assessment by day 7. Given the high inflammatory state associated with SAH and our intention to maintain a steady reduction in inflammation, we decided to extend the treatment duration to 20 minutes per session.”

Addorisio, Meghan E., et al. "Investigational treatment of rheumatoid arthritis with a vibrotactile device applied to the external ear." Bioelectronic Medicine 5 (2019): 1-11.

The use of an acute biomarker of response is very important. A bimodal response to taVNS has been previously shown in patients with atrial fibrillation (Kulkarni et al. JAHA 2021).

Thank you for this valuable insight and for bringing the study by Kulkarni et al. to our attention. Their study showed that the response to Low-Level Tragus Stimulation (LLTS) varied among patients with atrial fibrillation, which can be predicted by acute P-wave alternans (PWA) to some degree. We have discussed the implication of the bimodal response to taVNS in the Discussion section (p26-27): “Kulkarni et al. showed that the response to low-level tragus stimulation (LLTS) varied among patients with atrial fibrillation.49 Similarly, in our study, not all patients in the taVNS treatment group showed a reduction in mRS scores (improved degree of disability or dependence). This differential response may be inherent to taVNS and potentially influenced by factors such as anatomical variations in the distribution of the vagus nerve at the outer ear. These findings underscore the importance of using acute biomarkers to guide patient selection and optimize stimulation parameters. Furthermore, we found that increased heart rate was a potential acute biomarker for identifying SAH patients who are most likely to respond favorably to taVNS treatment. Translating this finding into clinical practice will require further research to elucidate the mechanisms by which an acute increase in heart rate may predict the outcomes of patients receiving taVNS, including its relationship with neurological evaluations, vasospasm, echocardiography, and inflammatory markers.”

**Reviewer #2 (Public Review):**
Summary:This study investigated the effects of transcutaneous auricular vagus nerve stimulation (taVNS) on cardiovascular dynamics in subarachnoid hemorrhage (SAH) patients. The researchers conducted a randomized clinical trial with 24 SAH patients, comparing taVNS treatment to a Sham treatment group (20 minutes per day twice a day during the ICU stay). They monitored electrocardiogram (ECG) readings and vital signs to assess acute as well as middle-term changes in heart rate, heart rate variability, QT interval, and blood pressure between the two groups. The results showed that repetitive taVNS did not significantly alter heart rate, corrected QT interval, blood pressure, or intracranial pressure. However, it increased overall heart rate variability and parasympathetic activity after 5-10 days of treatment compared to the sham treatment. Acute taVNS led to an increase in heart rate, blood pressure, and peripheral perfusion index without affecting corrected QT interval, intracranial pressure, or heart rate variability. The acute post-treatment elevation in heart rate was more pronounced in patients who showed clinical improvement. In conclusion, the study found that taVNS treatment did not cause adverse cardiovascular effects, suggesting it is a safe immunomodulatory treatment for SAH patients. The mild acute increase in heart rate post-treatment could potentially serve as a biomarker for identifying SAH patients who may benefit more from taVNS therapy.Strengths:The paper is overall well written, and the topic is of great interest. The methods are solid and the presented data are convincing.Weaknesses:(1) It should be clearly pointed out that the current paper is part of the NAVSaH trial (NCT04557618) and presents one of the secondary outcomes of that study while the declared first outcomes (change in the inflammatory cytokine TNF-α in plasma and cerebrospinal fluid between day 1 and day 13, rate of radiographic vasospasm, and rate of requirement for long-term CSF diversion via a ventricular shunt) are available as a pre-print and currently under review (doi: 10.1101/2024.04.29.24306598.). The authors should better stress this point as well as the potential association of the primary with the secondary outcomes.

Thank you for this valuable suggestion. The current study indeed focuses on the trial’s secondary outcomes. The main objective is to evaluate the cardiovascular safety of the taVNS protocol and to provide insights that will inform the application of taVNS in SAH patients. Following your comments, we have clarified this in the Introduction section (p6): “The current study is part of the NAVSaH trial (NCT04557618) and focuses on the trial’s secondary outcomes, including heart rate, QT interval, HRV, and blood pressure.32 This interim analysis aims to evaluate the cardiovascular safety of the taVNS protocol and to provide insights that will inform the application of taVNS in SAH patients. The primary outcomes of this trial, including change in the inflammatory cytokine TNF-α and rate of radiographic vasospasm, are available as a pre-print and currently under review.26”

The negative association between HRV and inflammatory cytokines has been reported in numerous studies such as (Williams et al., Brain, Behavior, and Immunity, 2019; Haensel et al., Psychoneuroendocrinology. 2008). There are some studies suggesting that increased sympathetic tone following SAH is associated with vasospasm (Bjerkne Wenneberg, S. et al., Acta Anaesthesiologica Scandinavica. 2020; Megjhani et al., Neurocrit Care. 2020). Based on the literature, we compared the effects of taVNS on primary and secondary outcomes. The findings from the two parallel analyses are consistent: taVNS treatment reduced pro-inflammatory cytokines and increased HRV. Furthermore, the analyses of the primary outcomes revealed a reduction in the presence of any radiographic vasospasm in the taVNS treatment group compared to the sham. We have now integrated these findings and discussed them in the Discussion section (p25-26): “Given the negative association between pro-inflammatory markers and HRV, our finding that HRV was higher in the taVNS treatment group aligns with the findings of primary outcomes of this clinical trial, which showed that taVNS treatment reduced pro-inflammatory cytokines, including tumor necrosis factor-alpha (TNF-α) and interleukin-6.26,52 The consistency between these findings strengthens the evidence supporting the anti-inflammatory effects of taVNS. In addition, the sympathetic predominance following SAH is implicated in an increased risk of delayed cerebral vasospasm, which is most commonly detected 5-7 days after SAH.12 Given that taVNS treatment mitigated the sympathetic overactivation before the typical onset of cerebral vasospasm, it could potentially reduce the severity of this complication.”

(2) The references should be implemented particularly concerning other relevant papers (including reviews and meta-analysis) of taVNS safety, particularly from a cardiovascular standpoint, such as doi: (10.1038/s41598-022-25864-1 and doi: 10.3389/fnins.2023.1227858).

Thank you for providing the relevant papers. We have provided these references in the Introduction section to provide a more comprehensive background of our study (p6): “While some animal studies have reported a potential risk of bradycardia and decreased blood pressure associated with vagus nerve stimulation, two reviews of human studies have considered the cardiovascular effects of taVNS generally safe, with adverse effects reported only in patients with pre-existing heart diseases. 21,22,23

(3) The dose-response issue that affects both VNS and taVNS applications in different settings should be mentioned (doi: 10.1093/eurheartjsupp/suac036.) as well as the need for more dose-finding preclinical as well as clinical studies in different settings (the best stimulation protocol is likely to be disease-specific).Overall, the present work has the important potential to further promote the usage of taVNS even on critically ill patients and might set the basis for future randomized studies in this setting

Thank you for this valuable insight. Scientific understanding of the dose-response relationship and determining optimal parameters tailored to specific disease contexts has been recognized as an important part of taVNS research and, more generally, in the electrical neuromodulation field. Studies in this direction are often complex and time-intensive due to the multitude of possible parameter combinations. As such, most taVNS studies opted to use parameters that have been established in previous studies. For example, 20 Hz taVNS is extensively used as a therapeutic intervention in stroke (Matyas Jelinek ,2024, https://www.sciencedirect.com/science/article/pii/S0014488623003138). As we pioneer the application of taVNS as an immunomodulation technique in SAH patients, we also adopt parameters reported in similar studies, aiming to provide a basis for future preclinical and clinical studies of taVNS in this patient population. As you noted, the effects of taVNS are dose-dependent, necessitating systematic exploration of the parameter space, including frequency, intensity, and duration. Our findings of the acute biomarker (heart rate) hold the promise of close-loop taVNS. We have now emphasized the importance of investigating how parameters/dose affect taVNS’s effects on immune function and cardiovascular function in SAH patients (p28): “As we pioneer the application of taVNS as an immunomodulation technique in SAH patients, we adopt parameters (20 Hz, 0.4 mA) reported in similar studies.55 The current study provides a basis for future preclinical and clinical studies of taVNS in this patient population. To build on our findings, a systematic evaluation of the relationship between parameters such as frequency, intensity, and duration and taVNS’s effects on the immune system and cardiovascular function is necessary to establish taVNS as an effective therapeutic option for SAH patients.56”

**Reviewer #2 (Recommendations For The Authors):**
The paper is overall well written, and the topic is of great interest. The reviewer has some major comments:(1) It should be clearly pointed out that the current paper is part of the NAVSaH trial and presents one of the secondary outcomes of that study while the declared first outcomes (change in the inflammatory cytokine TNF-α in plasma and cerebrospinal fluid between day 1 and day 13, rate of radiographic vasospasm, and rate of the requirement for long-term CSF diversion via a ventricular shunt) are available as a pre-print and currently under review (doi: 10.1101/2024.04.29.24306598.).

We have revised the manuscript following your comment. Please see comment Reviewer 2 Public Review and our response.

The authors should assess the relationship between the impact of taVNS on inflammatory markers in plasma and in cerebrospinal fluid and the autonomic responses. The association between inflammatory markers and noninvasive autonomic markers as well as sympathovagal balance should also be assessed. Specifically, the authors should try to assess whether the acute post-treatment elevation in heart rate was more pronounced in patients who experienced a more pronounced reduction in inflammatory biomarkers. Indeed, since all patients in the current study received the same dose of taVNS (20 Hz frequency, 250 μs pulse width, and 0.4 mA intensity), while in several cardiovascular studies (doi: 10.1016/j.jacep.2019.11.008, doi: 10.1007/s10286-023-00997-z) the intensity (amplitude) of taVNS was differentially set based on the subjective pain/sensory threshold, that might be a marker of acute afferent neuronal engagement.

We agree that analyzing the change in cardiovascular metrics and changes in inflammatory markers is an important next step. In particular, testing whether the acute elevation in heart rate correlates with changes in inflammatory markers could further establish heart rate as a biomarker to guide patient selection and optimize stimulation parameters. (Please refer to comment 1.3 and our responses). However, in this paper, the primary objective is the cardiovascular safety of the current taVNS protocol in SAH patients. This association between inflammatory markers and autonomic responses extends beyond the scope of the current manuscript and would be more appropriately addressed in a separate publication.

Previous literature has shown a negative association between HRV and inflammatory markers in SAH patients (for example, Adam, J., 2023). It is reasonable to postulate that taVNS modulates the immune system and the autonomic system synergistically. We found that parasympathetic tone was higher in the taVNS treatment group, with the most notable differences observed between Days 2 and 4 following SAH (Figure 3F). In a separate study of the primary outcomes of this trial (Huguenard et al., 2024), serum levels of IL-6 (pro-inflammation cytokine) were also significantly lower in the taVNS treatment group on Day 4 (Figure 3A, in our preprint, https://doi.org/10.1101/2024.04.29.24306598).

We appreciate your input regarding the potential mechanism behind acute heart rate changes. In this trial, all patients who were able to engage in verbal communication were asked if they felt any prickling or pain during all sessions. We confirmed that the current stimulation setting was sub-perception in all trialed patients, making it unlikely that the observed heart rate increase was due to pain or sensory perception. Our current hypothesis is that successful activation of the afferent vagal pathway by taVNS increased arousal, resulting in increased heart rate. We have revised the Discussion section based on your insight (p29): “All patients who were capable of verbal communication were asked if they felt any prickling or pain during all sessions. We confirmed that the current taVNS protocol is below the perception threshold for all trialed patients. Altogether, successful activation of the afferent vagal pathway by taVNS increased arousal, resulting in increased heart rate.50,51”

Huguenard, A. L. et al. Auricular Vagus Nerve Stimulation Mitigates Inflammation and Vasospasm in Subarachnoid Hemorrhage: A Randomized Trial. (2024) doi:10.1101/2024.04.29.24306598.

Adam, J., Rupprecht, S., Künstler, E. C. S. & Hoyer, D. Heart rate variability as a marker and predictor of inflammation, nosocomial infection, and sepsis – A systematic review. Autonomic Neuroscience vol. 249 103116 (2023).

A new table should be provided with the mean (or median) values of the two arms of the population (taVNS and sham) including baseline clinical characteristics, comorbidities (mean age, % of female, % with known hypertension, diabetes, etc), ongoing medications (% on beta-betablockers, etc), and pre, during and post-treatment absolute values (expressed as mean or median depending on the distribution) of the studied parameters (QT and QTc absolute values, heart rate, SDNN, etc) in order for the reader to have a better understanding of how SAH affects these parameters. Absolute changes in the abovementioned parameters should also be presented in the table. For instance, the reported absolute increase in heart rate, based on Figure 5, panel C, seems very modest, below 2 bpm. This is very important to underlying for several reasons, including the fact that the evaluation of the impact of treatment on heart rate variability as assessed in the time domain might be influenced by concomitant changes in heart rate due to the nonlinearity of neural modulation of sinus node cycle length. Indeed, time-domain indexes of HRV intrinsically increase when heart rate decreases in a nonlinear way, while frequency domain indexes (e.g. the low frequency/high frequency (LF/HF) ratio), appear to be devoid of intrinsic rate-dependency (doi: 10.1016/s0008-6363(01)00240-1).

Thank you for your suggestion. We have added the new table to the manuscript. In this table, we include clinical characteristics, the median of absolute values of cardiovascular metrics from 24-hour ECG recording, and the median absolute changes in these metrics for both arms. We believe that absolute values of cardiovascular metrics from 24-hour ECG recording are more informative about how SAH affects these parameters than metrics for the pre-, during-, and post-treatment periods.

In Result (p7), we have added: “Supplementary Table 3 shows the clinical characteristics of the two treatment groups.” In Result, Acute effect of taVNS on cardiovascular function (p20), we have added: “Supplementary Table 3 summarizes the absolute changes in cardiovascular metrics for the treatment groups.”

Thank you for raising the concern about HRV and providing the reference. We have now reported frequency domain indexes in our results: relative power of high-frequency power, which is negatively correlated with the LF/HF ratio. The high-frequency power is used to capture sinus arrhythmia, reflecting the parasympathetic modulation of the heart. Although the frequency domain metrics might be less susceptible to the rate-dependency (doi: 10.1016/s0008-6363(01)00240-1), there are circumstances when the frequency domain metrics might not accurately reflect the autonomic tone (Please see Reviewer 1 Publice Review and our responses).

An attempt to correct the effect of taVNS on the evaluated autonomic parameters according to age should be provided, considering that there were no age limits and parasympathetic indexes, particularly at the sinus node level, are known to decrease with age, particularly for those older than 65 years.

Thank you for the suggestion. We were aware of the influence of age on cardiac heart rate and heart rate variability. In our initial analysis, we compared the change in autonomic parameters from day 1 within each subject across the two treatment groups. This approach controls for individual differences, including those due to age. In addition to your comment, age is a risk factor for subarachnoid hemorrhage. Older individuals often face an increased risk of poor outcomes. To further verify if age influences autonomic changes following SAH, we performed ANCOVA on autonomic function parameters with age included as a covariate. This analysis showed that age was negatively correlated with changes in heart rate, SDNN, and RMSSD from Day 1, but not with changes in QT intervals. After adjusting for age, we found that RMSSD changes and SDNN changes were significantly higher in the taVNS treatment group, while QTc changes were significantly lower in this group. These results align with the main findings (Figures 2 and 3). In addition, autonomic changes following SAH may be influenced by age. Specifically, lower RMSSD and SDNN in older individuals suggest a greater shift toward sympathetic predominance following SAH. We have now reported these results in Supplementary Table 4 and discussed their implication in the Discussion section (p28): “To control for individual differences, including those due to age, our study compared the change in cardiovascular parameters from Day 1 within each subject across treatment groups. To further verify if age influences autonomic changes following SAH, we performed ANCOVA on autonomic function parameters with age included as a covariate. This analysis showed that age was negatively correlated with changes in heart rate, SDNN, and RMSSD from Day 1 but not with changes in QT intervals. After adjusting for age, we found that RMSSD changes and SDNN changes were significantly higher, while QTc changes were significantly lower in the taVNS treatment group (Supplementary Table 4). These results align with the conclusion that repetitive taVNS treatment increased HRV and was unlikely to cause bradycardia or QT prolongation. In addition, autonomic changes following SAH may be influenced by age. Specifically, lower RMSSD and SDNN in older individuals suggest a greater shift toward sympathetic predominance following SAH (Supplementary Table 4).”

The results of the current study should be discussed considering what was previously demonstrated concerning the cardiovascular effects of taVNS (doi: 10.3389/fnins.2023.1227858).

We appreciate the suggestion to consider previous findings on the cardiovascular effects of taVNS. However, it is important to note that most studies investigating the cardiovascular effects of taVNS involve healthy individuals, whereas our study focuses on SAH patients who are critically ill. Given the influence of SAH on cardiovascular parameters, we should be cautious when generalizing our findings to the broader population. Previous studies involving stroke populations have reported cardiovascular parameters descriptively as part of their safety assessments (doi: 10.1155/2020/8841752). Our study is currently the only one systematically investigating the cardiovascular safety of taVNS in SAH patients. Furthermore, the review paper (doi: 10.3389/fnins.2023.1227858) includes a highly heterogeneous mix of studies, such as auricular acupressure, auricular acupuncture, and electrical stimulation applied to different parts of the ear. For the subset of studies involving electrical stimulation, there is considerable variation in the parameters used, with frequencies ranging from 0.5 Hz to 100 Hz, currents from 0.1 mA to 45 mA, and durations spanning from 20 minutes to 168 days. These variations make direct comparisons with our findings challenging.

It looks like QT measurements were performed automatically. It should be specified which method was used for the measurements (threshold, tangent, or superimposed method?).

In our study, QT intervals were measured based on thresholding after wavelet transforming the ECG signals (Martínez, J. P., IEEE Transactions on Biomedical Engineering, 2004, doi: 10.1109/TBME.2003.821031). The local maxima of the wavelet transform correspond to significant changes in the ECG signal, such as the rapid upward or downward deflections associated with the QRS complex. The algorithm searches modulus maxima, that is, peaks of wavelet transform coefficients that exceed specific thresholds, to identify the QRS complex. R peaks are found as the zeros crossing between the positive-negative modulus maxima pair. After localizing the R peak, the Q onset is detected as the beginning of the first modulus maximum before the modulus maximum pair created by the R wave. To identify the T wave, the algorithm searches for local maxima in the absolute wavelet transform in a search window defined relative to the QRS complex. Thresholding is used to identify the offset of the T wave. Please refer to comments 3.4 and 3.5 and our responses for details. We have clarified the method for measuring QT in the Method section (p35): “This algorithm identifies the QRS complex by searching for modulus maxima, which are peaks in the wavelet transform coefficients that exceed specific thresholds. The onset of the QRS complex is determined as the beginning of the first modulus maximum before the modulus maximum pair created by the R wave. To identify the T wave, the algorithm searches for local maxima in the absolute wavelet transform in a search window defined relative to the QRS complex. Thresholding is used to identify the offset of the T wave.”

QTc dispersion was not evaluated, and this should be listed as a limitation of the current study.

We have added this limitation in the Discussion section: Limitations and outlook (p31): “The current study did not explore the effects of taVNS on less commonly used cardiovascular metrics, such as QTc dispersion.”

It has been recently suggested (doi: 10.1016/j.brs.2018.12.510) that QTc, as a potential indirect marker of HRV, might be used as a biomarker for VNS response in the treatment of resistant depression. The author should try to assess whether in the current study baseline QTc before taVNS is associated with outcome and with taVNS response.

Thank you for the suggestion. The conference abstract in the provided doi stated that QTc as an indirect marker of HRV before implantation was correlated with changes in the depression rating scale. The mechanism seems to be that QTc has information about the pathophysiology of the depression (10.1097/YCT.0000000000000684). The current study focused on the comparison between taVNS treatment and sham treatment. Our future study will further test if SAH patients’ response to taVNS can be predicted by baseline QTc.

The dose-response issue that affects both VNS and taVNS in different settings should be mentioned (doi: 10.1093/eurheartjsupp/suac036.) as well as the need for more dose-finding preclinical as well as clinical studies in different settings (the best stimulation protocol is likely to be disease-specific).

Please refer to our responses to comment 3.

Minor CommentsSome typos or commas instead of affirmative points and vice versa.

Thank you for pointing this out. We have carefully proofread the manuscript and made the necessary corrections to ensure proper punctuation and grammar throughout.

Table 1: why age is expressed as a range for each person?

MedRxiv asks authors to remove all identifying information. Precise ages are direct identifiers, as opposed to age ranges. We have now revised the age column to ‘decade of life’ in the updated table. We believe this modification reduces confusion while adhering to MedRxiv’s guidelines.

Although already reported in the study protocol (doi: 10.1101/2024.03.18.24304239), the heart rate limits for inclusion should be reported (sustained bradycardia on arrival with a heart rate < 50 beats per minute for > 5 minutes, implanted pacemaker or another electrical device).

We have now added the specific inclusion and exclusion criteria in the Method details section (p33): “Inclusion criteria were: (1) Patients with SAH confirmed by CT scan; (2) Age > 18; (3) Patients or their legally authorized representative are able to give consent. Exclusion criteria were: (1) Age < 18; (2) Use of immunosuppressive medications; (3) Receiving ongoing cancer therapy; (4) Implanted electrical device; (5) Sustained bradycardia on admission with a heart rate < 50 beats per minute for > 5 minutes; (6) Considered moribund/at risk of imminent death.”

Why did the authors choose a taVNS schedule of two times per day of 30 minutes each as compared for instance to one hour per day? Please comment on that also referring to other taVNS studies in the acute setting such as the one by Dasari T et al (doi: 10.1007/s10286-023-00997-z.) where taVNS was applied for 4 hours twice daily. For instance, Yum Kim et al (doi: 10.1038/s41598-022-25864-1) recently reported in a systematic review and meta-analysis of taVNS, safety, that repeated sessions and sessions lasting 60 min or more were shown to be more likely to lead to adverse events.The International Consensus-Based Review and Recommendations for Minimum Reporting Standards in Research on Transcutaneous Vagus Nerve Stimulation should be referred to and contextualized (doi: 10.3389/fnhum.2020.568051).

Thank you for raising this question and providing relevant references. We have reviewed the proposed checklist for minimum reporting items in taVNS research (10.3389/fnhum.2020.568051) and have ensured that our manuscript complies with the recommended reporting items.

The current taVNS schedule was based on findings from Addorisio et al. (2019). We have revised the manuscript to clarify the rationale behind the current taVNS protocol. Please refer to our response to comment 1.2. The two studies mentioned in the comments were published after our trial was designed and initiated (https://clinicaltrials.gov/study/NCT04557618). Based on the meta-analysis by Yum Kim et al., the short duration of treatment sessions might explain the cardiovascular safety of the current taVNS protocol. We are also currently assessing the effects of our taVNS protocol on inflammatory markers.

**Reviewer #3 (Public Review):**
Summary:The authors aimed to characterize the cardiovascular effects of acute and repetitive taVNS as an index of safety. The authors concluded that taVNS treatment did not induce adverse cardiovascular effects, such as bradycardia or QT prolongation.Strengths:This study has the potential to contribute important information about the clinical utility of taVNS as a safe immunomodulatory treatment approach for SAH patients.Weaknesses:A number of limitations were identified:(1) A primary hypothesis should be clearly stated. Even though the authors state the design is a randomized clinical trial, several aspects of the study appear to be exploratory. The method of randomization was not stated. I am assuming it is a forced randomization given the small sample size and approximately equal numbers in each arm.

Thank you for the suggestion. The current study is part of the NAVSaH trial (NCT04557618), aiming to define the effects of taVNS on inflammatory markers, vasospasm, hydrocephalus, and continuous physiology data. This study focuses on the effects of repetitive and acute taVNS on continuous physiology data to evaluate the cardiovascular safety of the current taVNS protocol. The primary hypothesis tested in our study is that repetitive taVNS increased HRV but did not cause bradycardia and QT prolongation. Following your comments, we have clarified this in the Introduction section (p6): “This interim analysis aims to evaluate the cardiovascular safety of the taVNS protocol and to provide insights that will inform the application of taVNS in SAH patients. The primary outcomes of this trial, including change in the inflammatory cytokine TNF-α and rate of radiographic vasospasm, are available as a pre-print and currently under review.26 Based on a meta-analysis, repeated sessions lasting 60 min or more are likely to lead to aversive effects; therefore, we hypothesized that repetitive taVNS increased HRV but did not cause bradycardia and QT prolongation.23”

(2) The authors "first investigated whether taVNS treatment induced bradycardia or QT prolongation, both potential adverse effects of vagus nerve stimulation. This analysis showed no significant differences in heart rate calculated from 24-hour ECG recording between groups." A justification should be provided for why a difference is expected from 20 minutes of taVNS over a period of 24 hours. Acute ECG changes are a concern for increasing arrhythmic risk, for example, due to cardiac electrical restitution properties.

A human study (Clancy, L. A. et al., Brain Stimulation, 2017, https://doi.org/10.1016/j.brs.2014.07.031) has found that 15-min taVNS led to reduced sympathetic activity measured by low-frequency/high-frequency (LF/HF) ratio. The sympathetic activity remained lower than baseline levels during the recovery period, suggesting potential long-term effects of taVNS on cardiovascular function. In addition, the repetitive taVNS treatment in this clinical trial was intended to maintain a steady low-inflammatory state. Given the potential life-threatening implications of bradycardia and QT prolongation in these critically ill patients, we deemed it crucial to evaluate heart rate and QT interval both acutely and from 24-hour ECG monitoring. We have now provided the justification in the Result section (p11): “A study has shown that 15 minutes of taVNS reduced sympathetic activity in healthy individuals, with effects that persist during the recovery period.33 This finding suggests that taVNS may exert long-term effects on cardiovascular function. Therefore, we investigated whether repetitive taVNS treatment affects heart rate and QT interval, key indicators of bradycardia or QT prolongation, using 24-hour ECG recording.”

An additional value of analyzing 24-hour ECG recording is that we can detect bradycardia or QT prolongation that happen outside the period of the stimulation, which could caused by repetitive taVNS. To this end, we reanalyzed the data and calculated the percentage of prolonged QT intervals using 500ms criterion (Giudicessi, J. R., Noseworthy, P. A. & Ackerman, M. J. The QT Interval. Circulation, 2019). When comparing the percentage of prolonged QT intervals between the treatment groups, we found that changes in prolonged QT intervals percentage from Day 1 were higher in the Sham group (Figure 3F, Mann–Whitney U test, N(taVNS) = 94, N(Sham)=95, p-value < 0.001, Cohen’s d = -0.72). We have now reported the results in the Result section (p11): “To ensure that repetitive taVNS did not lead to QT prolongation happening outside the period of stimulation, we calculated the percentage of prolonged QT intervals. Prolonged QT intervals were defined as corrected QT interval >= 500 ms. We found that changes in prolonged QT intervals percentage from Day 1 were higher in the Sham group (Figure 3F, Mann–Whitney U test, N(taVNS) = 94, N(Sham)=95, p-value < 0.001, Cohen’s d = -0.72).

The concern regarding acute ECG changes related to increased arrhythmic risk is valid. We have improved the reasoning behind analyzing acute ECG change, which now reads (p20): “Assessing the acute effect of taVNS on cardiovascular is crucial for its safe translation into clinical practice. We compared the acute change of heart rate, corrected QT interval, and heart rate variability between treatment groups, as abrupt changes in the pacing cycle may increase the risk of arrhythmias.”

(3) More rigorous evaluation is necessary to support the conclusion that taVNS did not change heart rate, HRV, QTc, etc. For example, shifts in peak frequencies of the high-frequency vs. low-frequency power may be effective at distinguishing the effects of taVNS. Further, compensatory sympathetic responses due to taVNS should be explored by quantifying the changes in the trajectory of these metrics during and following taVNS.

We appreciate your concerns regarding the potential effects on the autonomic system associated with taVNS treatment. We would like to clarify that the primary objective of our study was to evaluate the cardiovascular safety of the taVNS protocol in SAH, with a specific focus on detecting any acute changes in heart rate and QT interval. As you highlighted, such acute ECG changes are a concern for increasing arrhythmic risk. By directly studying the trend of heart rate, HRV, and QT over the acute treatment periods, we found no significant change in these metrics between treatment groups. In addition, these metrics remained within 0.5 standard deviations of their daily fluctuations during and following taVNS treatment (Figure 5 and Supplementary Figure 6). These findings support the conclusion that the current protocol is unlikely to cause cardiac complications.

In response to your suggestion to conduct a more rigorous analysis, particularly concerning peak frequencies within the high-frequency (HF) and low-frequency (LF) bands, we pursued this analysis to explore more nuanced effects of taVNS on the autonomic system. We compared the shifts in peak frequencies within these bands between the treatment groups and found no significant changes that would suggest a sympathetic or parasympathetic shift following acute taVNS.

In detail, we have made the following revisions following your comments:

(1) We have clarified the motivation behind studying the acute change of cardiac metrics following taVNS treatment – monitoring the cardiovascular safety of current taVNS protocol in SAH patients (p18): please refer to response to comment 3.2.

(2) We compared the peak frequencies of the high-frequency and low-frequency bands following taVNS. added the results in the supplementary materials:

We note that neurophysiology underlying peak frequencies has not been thoroughly studied in the literature compared to the LF-band power or HF-band power. Therefore, we report this result as an exploratory analysis.

(3) We have added the changes of QTc during and following taVNS in Figure 5 and showed that they were within 0.5 standard deviations of their daily fluctuations during and following taVNS treatment. We have now shown the changes of HRV during and following taVNS in Supplementary Figure 6 A-D. We added the change of high-frequency power following Reviewer #1’s comment 1.1. Overall, our results suggest that repetitive taVNS increased parasympathetic activities, while there is no evidence that acute taVNS significantly affected heart rate or QT.

(4) The authors do not state how the QT was corrected and at what range of heart rates. Because all forms of corrections are approximations, the actual QT data should be reported along with the corrected QT.

The corrected QT interval (QTc) estimates the QT interval at a standard heart rate of 60 bpm. In practice, we removed RR intervals outside of the 300 – 2000 ms range. Further, we removed ectopic beats, defined as RR intervals differing by more than 20% from the one proceeding. We used the Bazett formula to correct the QT intervals: QTc=QTRR. We have now clarified how QT was corrected in the Method section – Data processing (p35-36): “R-peaks were detected as local maxima in the QRS complexes. P-waves, T-waves, and QRS waves were delineated based on the wavelet transform (Figure 2A-C).34 RR intervals were preprocessed to exclude outliers, defined as RR intervals greater than 2 s or less than 300 ms. RR intervals with > 20% relative difference to the previous interval were considered ectopic beats and excluded from analyses. After preprocessing, RR intervals were used to calculate heart rate, heart rate variability, and corrected QT (QTc) based on Bazett's formula: QTc=QTRR.44 The corrected QT interval (QTc) estimates the QT interval at a standard heart rate of 60 bpm.”

We have reported the actual QT data in the Result section (p10 and p 19):” Moreover, changes in corrected QT interval from Day 1 were significantly higher in the Sham group compared to the taVNS group (Figure 3B, Mann–Whitney U test, N(taVNS) = 94, N(Sham)=95, p-value < 0.001, Cohen’s d = -0.57). Similarly, uncorrected QT intervals from Day 1 were higher in the Sham group (Supplementary Figure 10A, Cohen’s d = -0.42).”

“Supplementary Figure 10B-C shows the acute changes in uncorrected QT interval.”

(5) The QT extraction method needs to be more robust. For example, in Figure 2C, the baseline voltage of the ECG is shifting while the threshold appears to be fixed. If indeed the threshold is not dynamic and does not account for baseline fluctuations (e.g., due to impedance changes from respiration), then the measures of the QT intervals were likely inaccurate.

A robust method to estimate the QT interval is essential in our study. To this end, we used the state-of-the-art method to calculate QT intervals. We first applied a 0.5 Hz fifth-order high-pass Butterworth filter and a 60 Hz powerline filter on the ECG recording. The high-pass filtering is used to correct potential baseline fluctuations. Subsequently, a wavelet-based algorithm was used to delineate the QRS complex and T wave (Martínez, J. P., IEEE Transactions on Biomedical Engineering, 2004). In short, this algorithm identifies QRS based on modulus maxima of the wavelet transform of ECG signals. After localizing the R peak, the Q onset is detected as the beginning of the first modulus maximum before the modulus maximum pair created by the R wave. The detection is performed on wavelet transform at a small scale rather than on the original signal, minimizing the effect of baseline shift (see III Detection methods, (5), Cuiwei Li et al., IEEE TBME, 1995, Detection of ECG Characteristic Points Using Wavelet Transforms). T wave is detected similarly based on wavelet transform. Please refer to our response to comment 2.9.

Martínez, J. P., Almeida, R., Olmos, S., Rocha, A. P., & Laguna, P. (2004). A wavelet-based ECG delineator: evaluation on standard databases. IEEE Transactions on Biomedical Engineering, 51(4), 570-581.

In Figure 2C, the purple and green lines take the value of 1 at the QRS onset or the T wave offset; otherwise, 0, which might appear to be a threshold. We have now used verticle lines to denote the detected QRS onsets and T wave offsets. Please see below for a comparison of the annotation:

We have clarified the details of extracting QT intervals from ECG recordings in the Method section (p31): “To calculate cardiac metrics, we first applied a 0.5 Hz fifth-order high-pass Butterworth filter and a 60 Hz powerline filter on ECG data to reduce artifacts. 35 We detected QRS complexes based on the steepness of the absolute gradient of the ECG signal using the Neurokit2 software package.35 R-peaks were detected as local maxima in the QRS complexes. P waves, T waves, and QRS complexes were delineated based on the wavelet transform of the ECG signals proposed by Martinez J. P. et al. (Figure 2A-C).36 This algorithm identifies the QRS complex by searching for modulus maxima, which are peaks in the wavelet transform coefficients that exceed specific thresholds. The onset of the QRS complex is determined as the beginning of the first modulus maximum before the modulus maximum pair created by the R wave. To identify the T wave, the algorithm searches for local maxima in the absolute wavelet transform in a search window defined relative to the QRS complex. Thresholding is used to identify the offset of the T wave.”

We have modified Figure 2C for better clarity:

More statistical rigor is needed. For example, in Figure 2D, the change in heart rate for days 5-7, 8-10, and 11-13 is clearly a bimodal distribution and as such, should not be analyzed as a single distribution. Similarly, Figure 2E also shows a bimodal distribution. Without the QT data, it is unclear whether this is due to the application of the heart rate correction method.

Thank you for raising this concern. Several factors could contribute to the observed distribution of changes in heart rate for days 5-7, 8-10, and 11-13, as shown in Figure 2D. One such factor is the smaller sample size in the later days. The mean duration of hospitalization for the 24 subjects included in this study was 11.29 days, with a standard deviation of 6.43, respectively. Other factors include variations in medical history, SAH pathology, and clinical outcomes during hospitalization. Further analysis revealed that heart rate was lower in patients with improved mRS scores (Supplementary Figure 4B), suggesting that clinical outcomes might impact changes in heart rate. Understanding the association between cardiovascular metrics and clinical assessments, such as vasospasm and inflammation, could help decide whether future taVNS trials should control for these factors when evaluating the effects of taVNS on cardiovascular function. We are currently continuing to recruit SAH patients in this clinical trial, and we plan to perform such analyses in future studies.

In the manuscript, we reported the effect size between the treatment groups for days 5-7, 8-10, and 11-13. This should be interpreted in conjunction with the characteristics of the distribution. To provide a rigorous interpretation of our results, we have now discussed these considerations in the discussion section (p28): “We noticed a high variance of change in heart rate for days 5 – 7, 8 – 10, and 11 – 13 for both treatment groups (Figure 2D). This may be due to the small sample size in the later days, given that the mean duration of hospitalization for the 24 subjects included in this study was 11.3 days with a standard deviation of 6.4. Differences in medical history and clinical outcomes during hospitalization may also explain the variance of change in heart rate for the later days. For example. heart rate was lower in patients with improved mRS scores (Supplementary Figure 4B). Understanding the association between cardiovascular metrics and clinical assessments, such as vasospasm and inflammation, could help decide whether future taVNS trials should control for these factors when evaluating the effects of taVNS on cardiovascular function.”

To test our hypothesis that repetitive taVNS does not induce significant heart rate change, we performed a two-tailed equivalence test of heart rate change between the two treatment groups, including data from days 2-13 (Figure 2D, left panel). To verify the validity of this approach, we calculated the Bimodality Coefficient (BC) and performed the Dip Test for unimodality for the distribution of heart rate change for the two treatment groups. The Bimodality Coefficient (BC) is a measure that combines skewness and kurtosis to assess whether a distribution is bimodal or unimodal. A BC value greater than 0.555 typically indicates a bimodal distribution, whereas a BC value less than or equal to 0.555 suggests an unimodal distribution. The Dip Test is a statistical test that assesses the unimodality of a distribution. A non-significant p-value (p-value ≥ 0.05) indicates that the distribution is likely unimodal. This analysis suggests that the distributions of heart rate changes in both treatment groups (days 2 - 13) are unimodal (BC = 0.457 and p = 0.374 for the taVNS treatment group; BC = 0.421 and p = 0.656 for the sham treatment group). This finding provides justification for our statistical approaches.

Figure 3A shows a number of outliers. A SDNN range of 200 msec should raise concern for a non-sinus rhythm such as arrhythmia or artifact, instead of sinus arrhythmia. Moreover, Figure 3B shows that the Sham RMSSD data distribution is substantially skewed by the presence of at least 3 outliers, resulting in lower RMSSD values compared to taVNS. What types of artifact or arrhythmia discrimination did the authors employ to ensure the reported analysis is on sinus rhythm? The overall results seem to be driven by outliers.

Mild cardiac abnormalities are common in SAH patients. Therefore, change in cardiovascular metrics was expected to differ from healthy individuals, which makes studying the cardiovascular effect on taVNS extremely important in this context. Following your comment, we investigated whether the large SDNN change was due to arrhythmia or artifacts. Except for a single instance where one subject exhibited an SDNN change of 200 ms on a particular day, all other SDNN changes were less than 150 msec. We identified the subject and day associated with the largest SDNN change, which is Day 7. As shown in Author response image 1A and B, SDNN of this subject increased on day 7 while the heart rate (HR) of this subject decreased. Changes in HRV were inversely related to HR changes, suggesting shifts in sympathetic and parasympathetic tone. We checked the ECG recording and the extracted NN intervals (processed RR intervals) on that day. The NN intervals are more variate on day 7 compared to day 1 (Author response image 1C and D). To determine whether the significant variance observed between 5:01 am and 5:02 am was due to arrhythmia or artifacts, we closely examined the corresponding ECG signals (Author response image 1E and F). Based on our analysis, the elevated SDNN is unlikely to be attributed to artifacts.

Similarly, we identified the subjects and days corresponding to the most prominent RMSSD decrease in the sham treatment group. We verified the ECG quality for this subject and the accuracy of RR interval identification, and that there was no significant cardiovascular event during the subject’s stay in the ICU. Based on the inclusion and exclusion criteria defined in our protocol (Huguenard A et al.m PLOS ONE, 2024), we did not exclude these data from our analysis.

Huguenard A, Tan G, Johnson G, Adamek M, Coxon A, et al. (2024) Non-invasive Auricular Vagus nerve stimulation for Subarachnoid Hemorrhage (NAVSaH): Protocol for a prospective, triple-blinded, randomized controlled trial. PLOS ONE 19(8): e0301154. https://doi.org/10.1371/journal.pone.0301154

To ensure accurate inferences about sympathetic and parasympathetic tone from these cardiovascular metrics, we have rigorously refined our methodologies, including correcting RR intervals outliers, correcting ectopic peaks, using state-of-art algorithms to identify QRS complex, P wave, and T wave (please refer to response to comment 3.5), and performing factor analysis. In addition, no significant cardiac complications have been reported by the attending physicians for the subjects included in this study. Nonetheless, it is important to note that ECG patterns in patients with SAH differ from those in healthy individuals, potentially impacting the accuracy of R peak identification. For example, one identified R peak (out of 73) was Q peak (F in the above figure). The pathology associated with SAH complicates the precise calculation of cardiovascular metrics and the interpretation of the results. We are committed to continually improving our methodologies for assessing autonomic function in SAH patients. We have now discussed these limitations in the Discussion section (p31-32): “Mild cardiac abnormalities are common in SAH patients5, complicating the precise calculation of cardiovascular metrics from ECG signals and the interpretation of the results. Systematic verification of methods for calculating cardiovascular metrics to ensure their applicability in SAH patients is crucial.”

The above concern will also affect the power analysis, which was reported by authors to have been performed based on the t-test assuming the medium effect size, but the details of sample size calculations were not reported, e.g., X% power, t-test assumed Bonferroni correction in the power analysis, etc.

Thank you for raising this concern. The current study is part of the NAVSaH trial (NCT04557618), focusing on the trial’s secondary outcomes (Please refer to comment 2.1 and our responses). The main objective of this interim analysis is to evaluate the cardiovascular safety of the current taVNS protocol. Goal enrollment for the pilot NAVSaH trial is 50 patients, based on power calculations to detect significant differences in inflammatory cytokines, radiographic vasospasm, and chronic hydrocephalus. The detailed power analysis is described in the protocol (Huguenard A et al.m PLOS ONE, 2024):

“Under a 2-by-2 repeated measures design consisting of two groups of patients, each measured at two time points, our goal is to compare the change across time in the taVNS group to the change across time in the Sham group. Based upon previous work from Koopman et al. [67], we assume our study will observe 1.1 standardized inflammatory cytokines mean change difference between the two groups. Using a two-sided, two-sample t-test, assuming both time points have equal variance and there is a weak correlation (i.e., 0.15) between measurement pairs, a sample size of 25 in each group achieves at least 80% power to detect a standardized difference of 1.1 in mean changes, with a significance level (alpha) of 0.05 [68].

Based on our preliminary data, we assume this study will observe 25% and 55% severe vasospasm in the taVNS and Sham groups, respectively. Under a design with 2 repeated measurements (i.e., 2 raters), assuming a compound symmetry covariance structure with a Rho of 0.2, at a significance level (alpha) of 0.05, a sample size of 25 in each group achieves at least 80% power when the null proportion is 0.55, and the alternative proportion is 0.25 [69–71].

As previously described, LV et al. [8] studied the relationship between cytokine levels and clinical endpoints in SAH, including hydrocephalus. From their outcomes, we predict a needed enrollment of approximately 50 to detect these endpoints. From our own preliminary data, with an incidence of chronic hydrocephalus 0% in treated patients and 28.6% in control (despite grade of hemorrhage), alpha = 0.05 and power = 0.80, the projected sample size to capture that change is approximately 44 patients.”

In this study, we used power analysis to report the achieved power of insignificant findings. For example, a Mann-Whitney U test on heart rate change between the treatment groups revealed no significant differences. We then used power analysis to calculate the achieved power. We have added the details of power analysis in the Method section (p34): “We calculated the achieved power of tests on heart rate change between the treatment groups assuming a medium effect size (Cohen’s d of 0.5) and a Type I error probability (a) of 0.05. Given that the Mann-Whitney U test is a non-parametric counterpart to the t-test and that the asymptotic relative efficiency of the U test relative to the t-test is 0.95 with normal distributions, we estimated the achieved power based on the power of a two-sample t-test, which is 0.93. We have clarified this in the introduction section and in the method section (p6 and p38):

“The current study is part of the NAVSaH trial (NCT04557618) and focuses on the trial’s secondary outcomes, including heart rate, QT interval, HRV, and blood pressure.30 This interim analysis aims to evaluate the cardiovascular safety of the taVNS protocol and to provide insights that will inform the application of taVNS in SAH patients. The primary outcomes of this trial, including change in the inflammatory cytokine TNF-α and rate of radiographic vasospasm, are available as a pre-print and currently under review.24”

“In this study, we reported the statistical power achieved for tests that yielded non-significant results. The achieved power is calculated based on a two-sample t-test assuming a medium effect size (Cohen’s d of 0.5) and a Type I error probability (a) of 0.05.”

If the study was designed to show a cardiovascular effect, I am surprised that N=10 per group was considered to be sufficiently powered given the extensive reports in the literature on how HRV measures (except when pathologically low) vary within individuals. Moreover, HRV measures are especially susceptible to noise, artifacts, and outliers.If the study was designed to show a lack of cardiovascular effect (as the conclusions and introduction seem to suggest), then a several-fold larger sample size is warranted.

The primary goal of this study is to assess the cardiovascular safety of the current taVNS protocol in SAH patients (please refer to comments 2.1 and 3.8 and our responses). More specifically, we want to assess whether the current taVNS protocol is associated with bradycardia or QT prolongation. The data in this study included ECG signals and vital signals from 24 subjects recruited between 2021 and 2024. The total number of days in the ICU is 271 days, which corresponds to 542 taVNS/sham treatment sessions. These data allow us to detect significant cardiovascular effects of acute taVNS with high power. For example, the comparison of heart rate from pre- to post-treatment sessions between treatment groups had power > 99% (N1 = 188, N2 = 199, assuming 0.05 type I error probability, medium effect size two sample t-test).

To safely conclude that there is no significant cardiovascular effect of repetitive taVNS on any given day following SAH, we would need to perform statistical tests between treatment groups on Day 1, Day 2, and Day N. In this context, 64 subjects per treatment group are required to achieve 80% power assuming medium effect size and 0.05 type I error probability (two-sample t-test). We have acknowledged this limitation in the Discussion section. Thank you for raising this concern!

The results reported in this study treat each day as an independent sample for several reasons. First, heart rate and HRV metrics exhibited great daily variations (Figure in comment 3.7, for example). Their value on one day was not predictive of the metrics on another day, which could be due to medications, interventions, or individualized SAH recovery process during the patient’s stay in the ICU. Second, SAH patients in the ICU often experience rapid/daily changes in clinical status, including fluctuations in intracranial pressure, blood pressure, neurological status, and other vital signs. Also, the recovery process from SAH is highly individualized, with different patients exhibiting distinct trajectories of recovery or complications. Day-to-day cardiovascular function changes varied as the patient recovered or encountered setbacks. Moreover, we verified ECG signal quality, corrected outliers and artifacts in ECG processing, and employed a state-of-the-art QRS delineation method (Please refer to comment 3.5). All these ensure the accuracy of our reported results.

The revised Discussion section now reads (31): ” Our study considers each day as an independent sample for the following considerations: 1. heart rate and HRV metrics exhibited great daily variations. Their value on one day was not predictive of the metrics on another day, which could be due to medications, interventions, or individualized SAH recovery process during the patient’s stay in the ICU. 2. SAH patients in the ICU often experience daily changes in clinical status, including fluctuations in intracranial pressure, blood pressure, neurological status, and other vital signs. 3. Day-to-day cardiovascular function changes varied as the patient recovered or encountered setbacks. To conclusively establish that there is no significant cardiovascular effect of repetitive taVNS on any given day following SAH, we would need to perform statistical tests between treatment groups for each day. In this context, 64 subjects per treatment group are required to achieve 80% power assuming medium effect size and 0.05 type I error probability (two-sample t-test).”